# Ediacaran biozones identified with network analysis provide evidence for pulsed extinctions of early complex life

A.D. Muscente[1,11], Natalia Bykova [2], Thomas H. Boag[3], Luis A. Buatois[4], M. Gabriela Mángano[4], Ahmed Eleish[5], Anirudh Prabhu[5], Feifei Pan[5], Michael B. Meyer[6], James D. Schiffbauer [7,8], Peter Fox[5], Robert M. Hazen[9] & Andrew H. Knoll[1,10]

Rocks of Ediacaran age (~635–541 Ma) contain the oldest fossils of large, complex organisms and their behaviors. These fossils document developmental and ecological innovations, and suggest that extinctions helped to shape the trajectory of early animal evolution. Conventional methods divide Ediacaran macrofossil localities into taxonomically distinct clusters, which may represent evolutionary, environmental, or preservational variation. Here, we investigate these possibilities with network analysis of body and trace fossil occurrences. By partitioning multipartite networks of taxa, paleoenvironments, and geologic formations into community units, we distinguish between biostratigraphic zones and paleoenvironmentally restricted biotopes, and provide empirically robust and statistically significant evidence for a global, cosmopolitan assemblage unique to terminal Ediacaran strata. The assemblage is taxonomically depauperate but includes fossils of recognizable eumetazoans, which lived between two episodes of biotic turnover. These turnover events were the first major extinctions of complex life and paved the way for the Cambrian radiation of animals.

[1] Department of Earth and Planetary Sciences, Harvard University, Cambridge, MA 02138, USA. [2] Trofimuk Institute of Petroleum Geology and Geophysics, Siberian Branch Russian Academy of Sciences, Novosibirsk 630090, Russia. [3] Department of Geological Sciences, Stanford University, Stanford, CA 94305, USA. [4] Department of Geological Sciences, University of Saskatchewan, Saskatoon, SK S7n 5E2, Canada. [5] Department of Earth and Environmental Sciences, Rensselaer Polytechnic Institute, Jonsson-Rowland Science Center, 1W19, 110 8th Street, Troy, NY 12180, USA. [6] Earth and Environmental Science Program, Harrisburg University of Science and Technology, Harrisburg, PA 17101, USA. [7] Department of Geological Sciences, University of Missouri, Columbia, MO 65211, USA. [8] X-ray Microanalysis Core Facility, University of Missouri, Columbia, MO 65211, USA. [9] Geophysical Laboratory, Carnegie Institution for Science, 5251 Broad Branch Road, Washington, D.C 20015, USA. [10] Department of Organismic and Evolutionary Biology, Harvard University, Cambridge, MA 02138, USA. [11]Present address: Department of Geological Sciences, Jackson School of Geoscience, University of Texas at Austin, Austin, TX 78712, USA. Correspondence and requests for materials should be addressed to A.D.M. (email: a.d.muscente@utexas.edu)

Macroscopic fossils in Ediacaran rocks document a diverse array of morphologically complex, multicellular organisms[1–6] that lived between 600 and 541 million years ago (Ma), before the ~541 Ma Cambrian radiation of animals[7]. These fossils include carbonaceous compressions of seaweeds and possible metazoans[5,8], secondarily (authigenically and diagenetically) mineralized remains of tubicolous organisms interpreted as eumetazoans[6], skeletons produced by early biomineralizing[9,10] and agglutinating[11] metazoans, and traces[12–14] left by motile animals. The most distinctive Ediacaran remains, however, are casts, molds, and impressions of soft-bodied organisms preserved in siliciclastic and, less commonly, carbonate rocks[4]. These Ediacara-type fossils are rarely found in Phanerozoic rocks but occur in Ediacaran strata around the world, recording an enigmatic group of organisms, colloquially known as the 'Ediacara biota.' Although the phylogenetic affinities of these organisms are unresolved, the Ediacara biota probably includes stem- and early crown-group animals, along with possible representatives of extinct non-metazoan clades[7,15,16]. Regardless of their precise affinities, Ediacaran fossils document the advent of ecological tiering[3,17], metazoan locomotion[12,14,18,19], skeletal biomineralization[9,10], macroscopic predation[13,20], and other innovations that ultimately set the stage for Cambrian animal diversification[3].

Few Ediacaran genera occur in rocks of Cambrian or younger age[1]. Some works attribute this observation to changes in preservational environments across the Ediacaran–Cambrian transition[2,21]. However, due to emerging research[22,23], such explanations have fallen out of favor in lieu of others citing biotic turnover. A large magnitude negative carbon-isotopic excursion near the Ediacaran–Cambrian boundary suggests that the extinction of Ediacaran taxa may have been driven, at least in part, by global environmental disturbance[24,25]. Additionally, the rise of eumetazoans may have contributed to one or more earlier episodes of ecological reorganization[1,2,6,23,26,27]. Detailed studies of deposits containing the Ediacaran–Cambrian boundary[23,25,28] provide empirical evidence that terminal Ediacaran strata are taxonomically depauperate relative to older Ediacaran and younger Cambrian rocks. If this observation represents a real signal, the Ediacara biota experienced an extinction prior to the end of the period, perhaps the first mass extinction of complex life in Earth history[2,23,27]. The evidence for this extinction, however, is based on analyses of individual sections and localities[23,25,28], which may not provide representative samples of the ancient biosphere[1]. In this context, hypothesis testing in Ediacaran paleobiology necessitates a robust biostratigraphic framework, allowing for discovery of global scale patterns in diversity over geologic time.

The chronology of evolutionary events at the dawn of animal life remains unclear due to uncertainties in the correlation of Ediacaran rocks. Unlike the younger systems of the geologic record, the Ediacaran has not yet been formally subdivided into chronostratigraphic units (i.e., series or stages). Many stratigraphic boundaries in the Phanerozoic Eon are defined by the first appearances of index fossils—abundantly preserved, morphologically distinct, and easily recognizable species with cosmopolitan distributions. However, most Ediacaran macrofossils represent soft-bodied organisms, and owing to environmental restrictions, taphonomic biases, and limited occurrences, their first appearances may vary among localities[29]. Consequently, efforts at correlation have largely concentrated on radiometric dating, microfossils[30], shelly fossils[9], and carbon-isotopic excursions, such as the potentially global Shuram anomaly[29]. Where both occur in the same section, morphologically complex microfossils generally occur below beds containing the Ediacara biota[5,30], and so, cannot be used to discriminate ages within the biota itself. The same is true of the Shuram excursion[29]. Shelly fossils, in contrast, occur in sedimentary successions containing non-skeletal macrofossils in China, Namibia, and the western US, among other places[9]. Indeed, the first appearance of the biomineralized fossil Cloudina, known from numerous localities around the world, has received attention as a potential marker of terminal Ediacaran strata[9,29]. Even so, shelly fossil taxa are generally preserved in carbonate rocks, limiting their application to siliciclastic-dominated successions. Given these and other issues, the International Subcommission on Ediacaran Stratigraphy has highlighted the need for innovative approaches to correlating the system[29].

Our work approaches the issue by seeking to identify biostratigraphic zones (biozones) based on fossil assemblages (associations). An assemblage biozone, in essence, is a set of lithologic packages representing roughly age-equivalent strata and the association of taxa preserved therein (Supplementary Table 1). Several prior investigations explored for assemblages in Ediacaran rocks[31–35]. Parsimony analysis[32], non-metric multidimensional scaling (NMDS)[31], and hierarchical clustering[31] all show that Ediacaran fossil localities can be sorted into three taxonomically distinct sets:[1,26] the Avalon, White Sea, and Nama clusters. These clusters are usually called 'assemblages,' but this terminology is problematic as it has multiple implications (e.g., life, death, and biozone assemblages). We consider the clusters to be paleocommunities—recurrent and distinguishable associations of taxa[36]—and herein, restrict our usage of 'assemblage' to the discussion of biozones and their characteristic fossil associations. Avalon localities include deposits in Newfoundland, northwestern Canada, and the United Kingdom, where relatively deep (marginal slope and basinal) turbiditic successions contain Ediacara-type fossils, including arboreomorph and rangeomorph fronds[17]. White Sea localities primarily occur in South Australia and Russia, where offshore and shoreface facies host diverse associations of Ediacara-type fossils, including many dickinsoniomorph, bilateralomorph, and kimberellomorph taxa[3,35]. Nama localities, in contrast, stand out for their taxonomically depauperate deposits with regard to Ediacara-type fossils, and correspond to shoreface and offshore facies with skeletal, trace, carbonaceous, and Ediacara-type fossils as well as conspicuous tubicolous elements[6]. They are located in Namibia, South China, and the western US, amongst other places. Some analyses[31] support the existence of a fourth cluster, herein referred to as the Miaohe cluster, named after shales with abundant macrofossils found near Miaohe, South China[5]. Localities in this cluster generally contain macroscopic carbonaceous compressions in fine-grained siliciclastic beds[5]. In any case, because communities vary through both time and space[36], the clusters remain a subject of debate[1,31]. Some authors have treated them as assemblage biozones[1,27,32,34], whereas others have argued that they represent biotopes—associations of taxa that inhabited environments with distinct boundaries, limited spatial overlap, and few shared species[1,31,33,35]. These interpretations are not mutually exclusive, and the variation may additionally reflect biases in fossil preservation[22].

To contribute to Ediacaran biostratigraphy and to test the hypothesis that macroscopic life experienced two extinctions during the Ediacaran–Cambrian transition, we herein explore data on occurrences of body and trace fossils in Ediacaran and lower Cambrian (Fortunian) rocks around the world (Methods). The dataset also includes information on morphogroups (see Supplementary Discussion), preservational modes (Supplementary Table 2), ichnofossil architectural designs (Supplementary Table 3), and paleoenvironments (Supplementary Table 4). We studied the data using conventional methods of paleoecological analysis[31], including hierarchical clustering and NMDS, as

well as an innovative network-based methodology. Network analysis encompasses an array of mathematical and visualization methods for examining complex systems[37]. Such methods have been employed to great effect in social science[38], biochemistry[39], proteomics[40], mineralogy[41], evolution[42], ecology[43], paleogeography[44], and paleoecology[36]. Our work applies network analysis to biostratigraphy and the study of biofacies. Using network-based methods, we integrate data on fossil occurrences with metadata on facies and stratigraphy, and thereby, distinguish between Ediacaran biotopes and biozones. Through this work, we provide empirically robust and statistically significant evidence for a global, cosmopolitan assemblage unique to terminal Ediacaran strata. This assemblage, which allows for the correlation of terminal Ediacaran rocks around the world, lived between two episodes of biotic turnover. These turnover events may represent the earliest mass extinctions of complex life. Regardless, the results corroborate the hypothesis that two extinctions helped to shape the trajectory of early animal evolution.

## Results

**Conventional methods**. Hierarchical clustering provides evidence for four groups of Ediacaran formations with five or more genera and ichnogenera (Fig. 1, Supplementary Figs. 1–4). These groups include the Avalon, White Sea, and Nama clusters as well as a fourth (Miaohe) set of formations. The Lantian Formation of China, which contains the early Ediacaran Lantian biota[8], falls outside of these clusters, regardless of the taxonomic similarity index used in clustering. All clusters identified with the Kulczynski-2 index are supported by approximately unbiased (AU) *P*-values > 95% (Fig. 1), indicating that those clusters are strongly supported by the data. Clusters identified with the Jaccard index are also supported by AU *P*-values > 95% (Supplementary Fig. 2), except the Nama cluster, which is supported at a lower but non-negligible (90%) confidence level. The NMDS analyses confirm that the clusters identified with hierarchical clustering are taxonomically distinct (Supplementary Figs. 3 and 4). Formations were ordinated, based on taxonomic similarity, in three-dimensional ordination space. A stress value, measuring the degree that ordination deviates from observed distances, was calculated for each NMDS analysis. These values were generally low (~0.10) for three-axis NMDS plots, indicating that such plots provide good representations of the formations' rank orders in reduced dimensions (Supplementary Fig. 3 and Supplementary Table 5). The plots show that the four clusters are distinct in ordination space, and the 95% confidence intervals around their centroids (mean scores) do not overlap, so the differences are statistically significant.

**Network analysis**. In general, network theory deals with the study of independent but interconnected entities[37]. A network consists of two basic elements: nodes (entities) and links (connections)[37]. The nodes in this study are taxa (body and trace fossil genera), paleoenvironments, and geologic formations, and the links among them represent connections supported by fossil occurrence data (e.g., links between taxa and their geologic formations and facies of occurrence). Network theory supports a diverse array of methods for building networked data structures and investigating their community units. This study includes unipartite and bipartite networks (Table 1, Supplementary Table 1)[37]. A unipartite network includes a single set of nodes, in which any pair of nodes may be connected. Each bipartite network, on the other hand, includes two sets of nodes, and a node may only be connected to another of a different set, so that nodes of the same set are never directly linked. This bipartite structure can be reduced to two unipartite network-like projections, one for each set of entities.

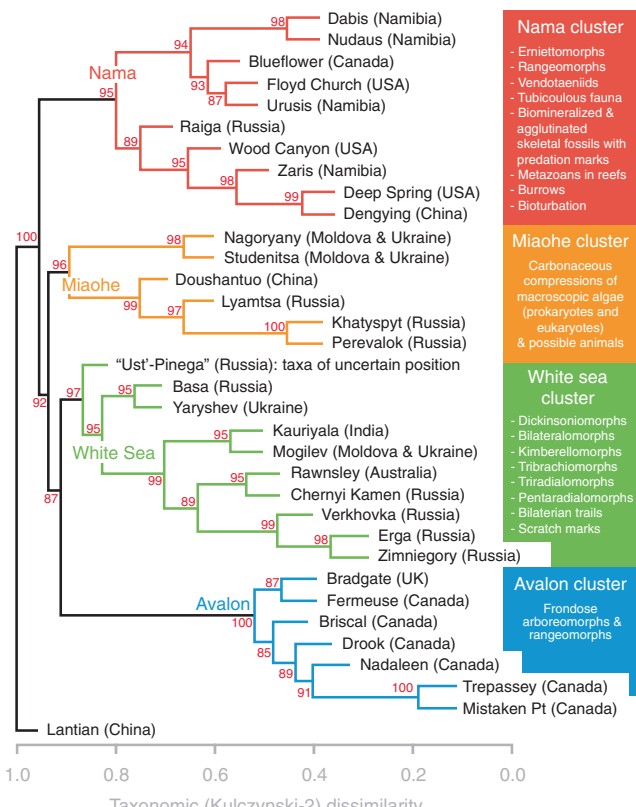

**Fig. 1** Hierarchical clustering. Dendrogram of 34 geologic formations containing five or more macrofossil genera and/or ichnogenera of Ediacaran age. Formations were clustered based on their taxonomic (Kulczynski-2) dissimilarities (*x*-axis) using the average-linkage method. The high cophenetic correlation score of the dendrogram (0.8506276)—the correlation between observed taxonomic dissimilarities and those estimated via hierarchical clustering—indicates that the dendrogram faithfully preserves the pairwise distances between the original data points. Red numbers are approximately unbiased (AU) *P*-values calculated for nodes from 1000 multiscale bootstrap resamples. From bottom to top above the Lantian Formation, the labeled Avalon (blue), White Sea (green), Miaohe (orange), and Nama (red) clusters have AU *P*-values ≥ 0.95 and are supported by the data. Inset boxes are lists of the clusters' dominant fossils. The "Ust'-Pinega" formation encompasses occurrences of taxa that belong to the Lyamtsa, Verkhovka, and Zimniegory formations but have uncertain stratigraphic positions. See Supplementary Information for Jaccard dissimilarity results (Supplementary Fig. 2). Source data are provided as a Source Data file

This procedure involves extrapolating direct links between nodes of one set from their indirect connections across nodes of another. In all networks, nodes and links make up community structures[37]. These structures consist of units (modules), which appear in network graphs as clusters of densely-connected nodes. Networks may contain multiple levels of community structure (e.g., clusters of communities), and the modules themselves may represent non-overlapping (mutually exclusive) groups or overlapping groups that share nodes.

Network analysis has several advantages over conventional methodologies used in paleoecology and biostratigraphy. First, it allows for the application of community-detection algorithms, which can be used to partition fossil networks into community-like modules[36,44,45] (Supplementary Table 1). Second, network theory accommodates an assortment of metrics for describing local (node specific) and global (whole network) properties and

**Table 1 List of networks in this study**

| Format | Node type 1 | Node type 2 (if present) | Description of links | Figure |
|---|---|---|---|---|
| Unipartite | Genera | – | Genera are connected to other taxa preserved at their collection points | Fig. 2a |
| Bipartite | Environments | Genera | Taxa are connected to their habitats and preservational environments | Fig. 3a |
| Bipartite | Environments | Genera & ichnogenera | Taxa are connected to their habitats and preservational environments | Supplementary Fig. 10a |
| Bipartite | Ediacaran formations | Genera & ichnogenera | Taxa are connected to formations where their fossils are preserved | Fig. 4a |
| Bipartite | Ediacaran formations | Genera & ichnogenera (shallow biotope) | Taxa are connected to formations where their fossils are preserved | Supplementary Fig. 11a |
| Bipartite | Ediacaran formations | Genera (shallow biotope) | Taxa are connected to formations where their fossils are preserved | Supplementary Fig. 12a |
| Bipartite | Ediacaran-Fortunian formations | Genera & ichnogenera | Taxa are connected to formations where their fossils are preserved | Supplementary Fig. 13a |

Each row is a network consisting of nodes and links. The nodes in this study are body macrofossil genera, trace fossil ichnogenera, paleoenvironments, and geologic formations. All valid taxa, which might serve as index fossils of ecological biotopes and geological biozones, were included in the networks (see Methods). See Supplementary Information for additional figures (Suppl. Figs.)

the variables underlying their community structures. Lastly, network theory supports analyses of integrated data structures that leverage multiple types of information. Partitioning of a bipartite network, for example, can lead to the discovery of community units based on the union of two sets of entities. Altogether, such methods indicate that network analysis can be used to discover, identify, and characterize ecologically and geologically meaningful associations of Ediacaran macrofossils.

In this study, we analyze one unipartite network and six bipartite networks derived from fossil occurrence data (Table 1). All of the networks contain taxa nodes (see Methods). In the unipartite network, two genera are connected if they have been reported together at one or more fossil collection points (e.g., beds, localities, sections, outcrops, etc.) anywhere in the world[36]. This structure represents the most basic expression of our dataset, and its modules generally represent paleocommunities[36]. The bipartite networks integrate occurrence data with metadata on facies (rock types corresponding to specific environments) and geologic units. In bipartite networks containing paleoenvironments, taxa are connected to their habitats and preservational environments[33,35] (Supplementary Table 4), and in bipartite networks containing formations, taxa are connected to geologic units where their fossils are preserved. In this context, the bipartite networks support investigation of Ediacaran biotopes and biozones, as their modules reflect unions of taxa with environments and geologic units. All other differences among the networks are related to raw data (Table 1).

We applied 14 community-detection algorithms to the networks to explore their community structures and identify paleocommunities, biotopes, and biozones (see Supplementary Discussion). Additionally, we calculated a number of metrics: modularity, homophily, and centrality (Supplementary Table 1). Modularity ($Q$) is a global property describing community structure. In its simplest form[46,47], the modularity of a community structure equals the fraction of links that connect nodes of the same communities minus the corresponding fraction expected in an equivalent network with a random distribution of connections. Homophily, another global property, measures the tendency of nodes to connect to others possessing similar nominal or continuous properties, and is measured with assortativity coefficients, which are similar to Pearson correlation coefficients. We calculated assortativity coefficients for the various network projections to assess the relative extents that their topologies reflect the underlying properties of their nodes,

such as the preservational modes[22] (Supplementary Table 2) and morphogroups[7,48] (see Supplementary Discussion and Supplementary Table 3) of the taxa and the locations of the formations. Finally, centrality represents a local property related to the relative importance of a node. A centrality score may, for example, equal a node's degree (number of links) or its betweenness, the number of shortest paths that pass through it (i.e., how often it serves as a bridge between other nodes).

Application of community-detection algorithms to the networks resulted in the discovery of numerous modules (Figs. 2–4; Supplementary Figs. 5–12). The community overlap propagation algorithm (COPRA) generally returned non-overlapping community structures with the fewest modules and greatest $Q$ scores (Supplementary Figs. 5–7 and Supplementary Table 6). It also returned overlapping modules, when procedures were formulated to allow taxa to be assigned to multiple community units. The overlapping community structures generally resemble their non-overlapping counterparts, except where nodes are assigned to multiple modules. Sensitivity analysis shows that the network partitioning results do not significantly change, even when at least 10% of connections (30–100 links) and five weakly connected (i.e., uncommon) taxa are omitted from each network (Supplementary Fig. 8). Moreover, data randomization indicates that it is unlikely that any the community structures reported in this study arose due to chance[49] (Supplementary Fig. 9). Thus, overall, the community structures are robust.

Application of the COPRA method to the unipartite network of Ediacaran genera (Fig. 2a) resulted in detection of four overlapping modules (Fig. 2b). The largest and most central cluster—the White Sea module—predominantly consists of bilateralomorph, dickinsoniomorph, and kimberellomorph genera, along with other Ediacara-type taxa (Fig. 2c, d). This cluster overlaps with the Avalon and Nama modules. Whereas the Avalon module primarily consists of rangeomorphs, the Nama cluster includes a mixture of taxa known from skeletal fossils, secondarily mineralized tubes, Ediacara-type fossils, and carbonaceous compressions. The remaining cluster, the Miaohe module, overlaps with the Nama module but is dominated by filament-, strap-, and ribbon-shaped taxa typically preserved as carbonaceous compressions. Assortativity coefficients indicate the best predictor of association in this network is preservational mode (Fig. 2c; Supplementary Fig. 7 and Supplementary Table 8). The relationship between morphology (form or morphogroup) and linkage is weak.

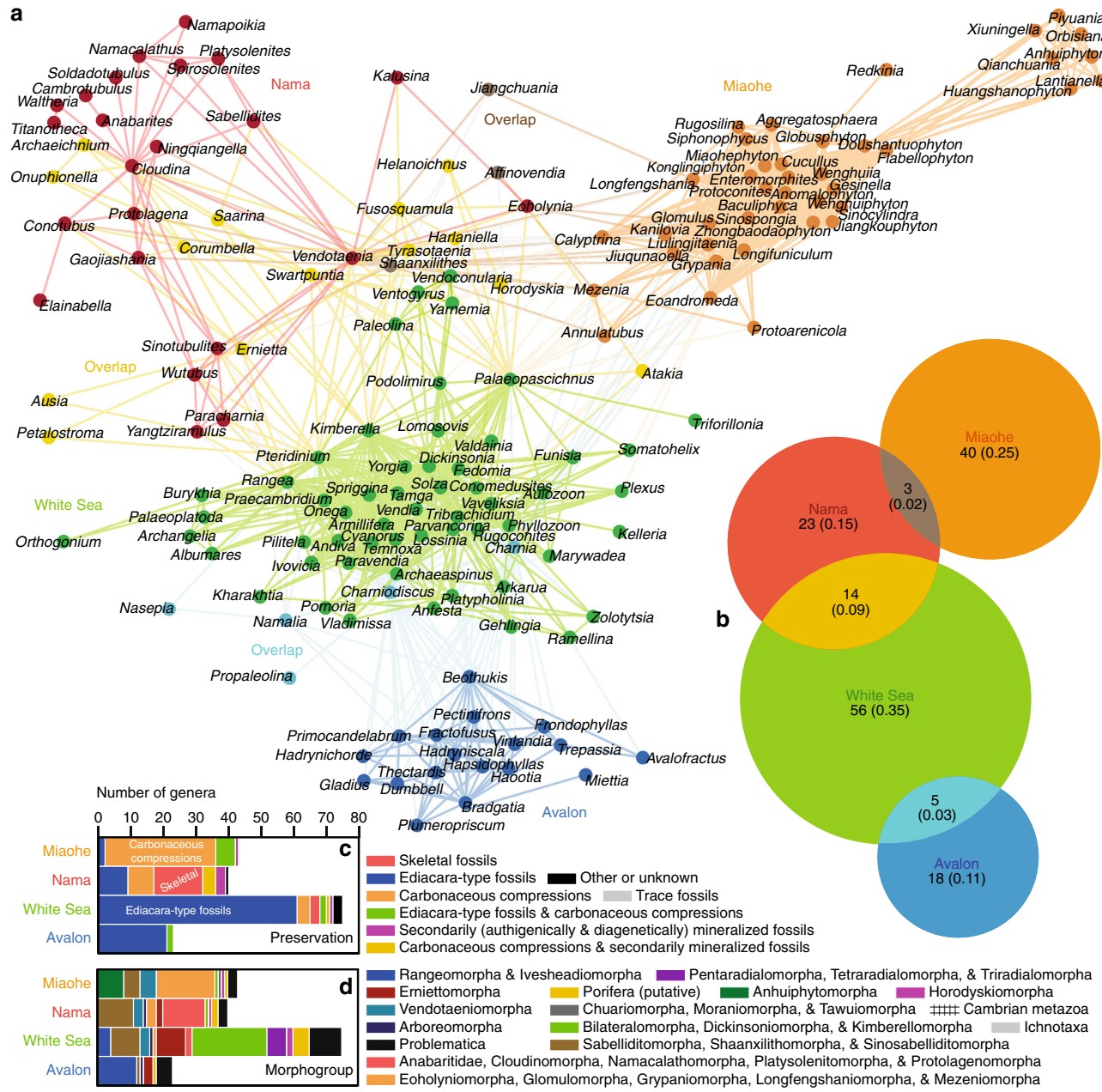

**Fig. 2** Unipartite network of Ediacaran macrofossil genera. **a** Network graph. Two genera are linked if those taxa cooccur at any fossil collection point in the dataset. Colors indicate modules identified using the COPRA community-detection algorithm ($v = 5$). According to randomization testing, this community structure is statistically significant (Supplementary Fig. 9; $Q = 0.82$, $P < 0.01$, $Z = 7.700$). All genera fall into four modules: the Avalon (blue), White Sea (green), Miaohe (orange), and Nama (red) clusters (Fig. 1; Supplementary Figs. 2–4). **b** Venn diagram illustrating module overlap. Areas of circles correspond to their relative numbers of genera, numbers are counts of taxa, and values in parentheses are proportions. **c, d** Stacked bar graphs showing numbers of genera in the modules and their preservational modes (**c**) and morphogroups (**d**). Source data are provided as a Source Data file

The COPRA method facilitated the detection of three modules in the bipartite network of paleoenvironments and Ediacaran genera (Fig. 3a, b). Each module includes one or more paleoenvironments and many genera, and therefore, resembles a biotope. The smallest module—the deep biotope—consists of Avalon taxa (Figs. 2a, 3c) known exclusively from turbiditic facies representing deep-water paleoenvironments (Supplementary Table 4). The next smallest module—the shallow biotope—predominately consists of White Sea and Nama genera, which occur in clastic and carbonate facies reflecting shallow water paleoenvironments regularly influenced by wave action and tidal processes (Supplementary Table 4). Conversely, the largest

module—the intermediate biotope (intermediate with regard to water depth)—is comprised of White Sea, Nama, and Miaohe taxa (Figs. 2a, 3c). These genera occur in clastic and carbonate facies representing offshore shelf and ramp paleoenvironments located below fair-weather wave base and characterized by low-energy conditions (Supplementary Table 4). This module also includes the carbonate slope and basin paleoenvironment, which shares various taxa with the offshore shelf, offshore shelf transition, and outer ramp paleoenvironments. Virtually all Miaohe genera (Fig. 2a) belong to this intermediate biotope (Fig. 3c). All three clusters share genera with each other, but the greatest overlap occurs between the shallow and intermediate

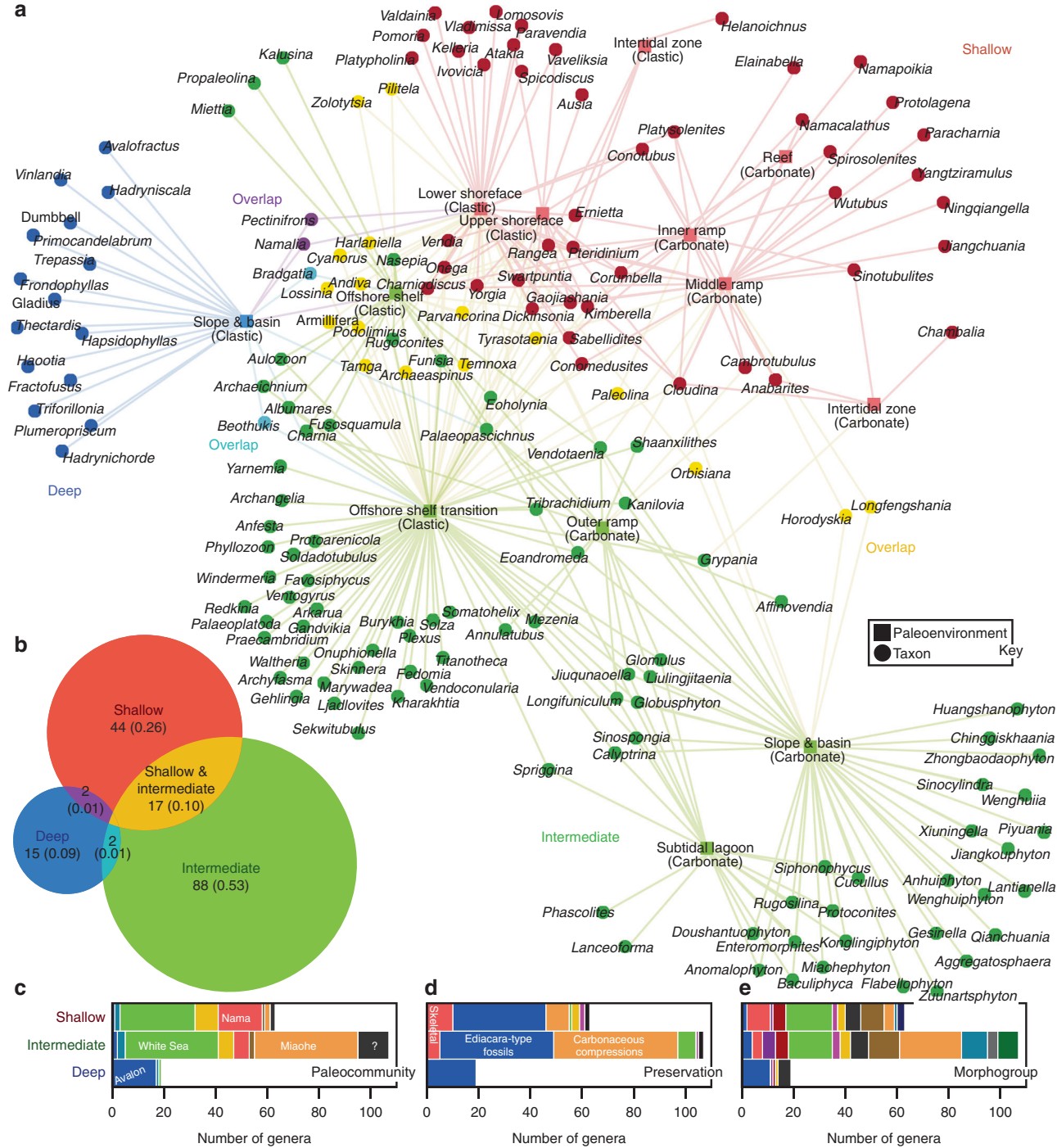

**Fig. 3** Bipartite network of paleoenvironments and Ediacaran macrofossil genera. **a** Network graph. A genus and paleoenvironment are linked if fossils of the taxon have been reported from matching facies (Supplementary Table 4). Colors indicate modules identified using the COPRA community-detection algorithm ($v = 2$). According to randomization testing, this community structure is statistically significant (Supplementary Fig. 9, paleoenvironments projection, $Q = 0.39$, $P = 0.05$, $Z = 1.46$; taxa projection, $Q = 0.51$, $P = 0.05$, $Z = 1.27$). All paleoenvironments and genera fall into three modules—the deep (blue), shallow (red), and intermediate (green) clusters—named for the relative water depths of their centroids along a model shallow-to-deep-water transect. **b** Venn diagram illustrating taxonomic overlap of modules. Areas of circles correspond to their relative numbers of genera, numbers are counts of taxa, and values in parentheses are proportions. **c** Stacked bar graph showing numbers of genera belonging to the various Ediacaran paleocommunities in each module (colors are those used in Fig. 2a, b). **d, e** Stacked bar graphs showing numbers of genera in the modules and their preservational modes (**d**) and morphogroups (**e**). See Fig. 2 for color keys and Supplementary Fig. 10 for related analyses. Source data are provided as a Source Data file

biotopes. Assortativity coefficients show that none of the nominal properties of the genera (e.g., preservational mode or morphogroup) represent strong predictors of association (Supplementary Fig. 7). Addition of ichnogenera to this network does

not significantly alter its community structure or the module assignments of taxa (Supplementary Fig. 10).

Partitioning of the bipartite network of Ediacaran formations and taxa (genera and ichnogenera) with COPRA again resulted in the

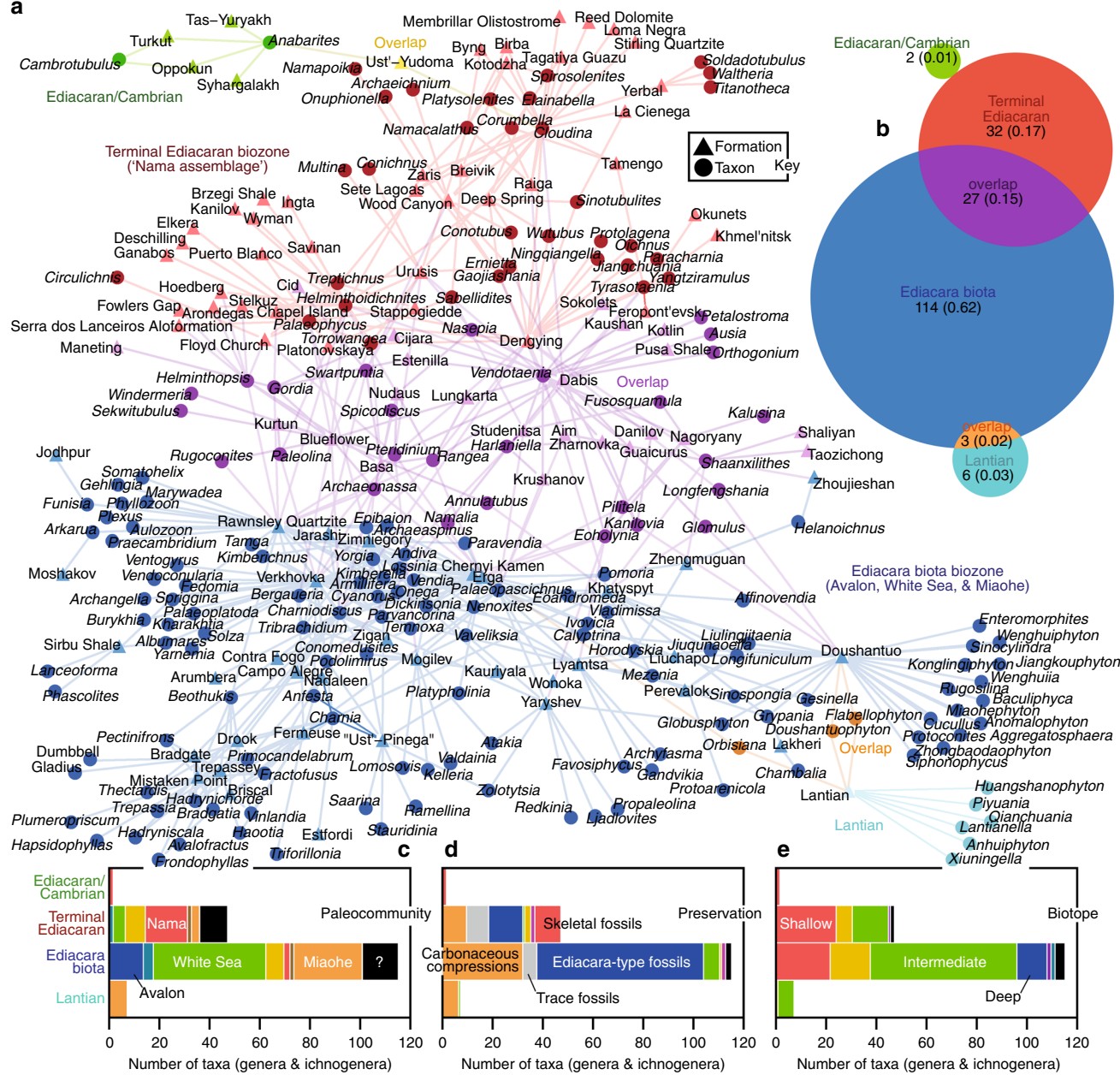

**Fig. 4** Bipartite network of Ediacaran formations and macrofossil taxa. **a** Network graph. A taxon (genus or ichnogenus) and geologic formation are linked if fossils of the taxon have been reported from that geologic unit. Colors indicate modules identified using the COPRA community-detection algorithm ($v = 6$). According to randomization testing, this community structure is statistically significant (Supplementary Fig. 9; formations projection, $Q = 0.63$, $P < 0.01$, $Z = 12.17$; taxa projection, $Q = 0.51$, $P < 0.01$, $Z = 6.57$). All formations and taxa fall into four modules: the Lantian biota (teal), Ediacara biota biozone (blue), Terminal Ediacaran biozone (red), and Ediacaran/Cambrian taxa (green) clusters. **b** Venn diagram illustrating taxonomic overlap of modules. Areas of circles correspond to their relative numbers of genera and ichnogenera, numbers are counts of taxa, and values in parentheses are proportions. **c** Stacked bar graph showing the numbers of taxa belonging to the various Ediacaran macrofossil paleocommunities (colors are those used in Fig. 2a, b) and representing traces in the modules. **d** Stacked bar graph showing numbers of taxa in the modules and their preservational modes (see Fig. 2 for color key). **e** Stacked bar graph showing numbers of taxa in the modules and their biotope assignments (colors are those used in Fig. 3a, b). See Supplementary Figs. 11 and 12 for related analyses. Source data are provided as a Source Data file

discovery of four overlapping modules (Fig. 4a, b). These modules resemble assemblage-based biozones (Fig. 5), in that they are defined by various taxa and strata (see Supplementary Discussion). Two modules contain the majority of nodes (Fig. 4b) as well as all nodes with high centrality scores (Fig. 6). The largest module—the Ediacara biota biozone (EBB)—is comprised of formations containing Avalon, White Sea, and Miaohe genera (Fig. 4c) known from Ediacara-type fossils and carbonaceous compressions (Fig. 4d).

Collectively, the taxa of the EBB represent all biotopes, with the majority belonging to the module of intermediate water depth (Figs. 3a, 4e; Supplementary Fig. 10a). In contrast, the second largest module—the Terminal Ediacaran biozone (TEB)—consists of formations dominated by Nama genera (Fig. 4c), including skeletal taxa and forms known from Ediacara-type fossils and carbonaceous compressions (Fig. 4d). These fossils are preserved in both nearshore and offshore settings (Figs. 3a, 4e; Supplementary

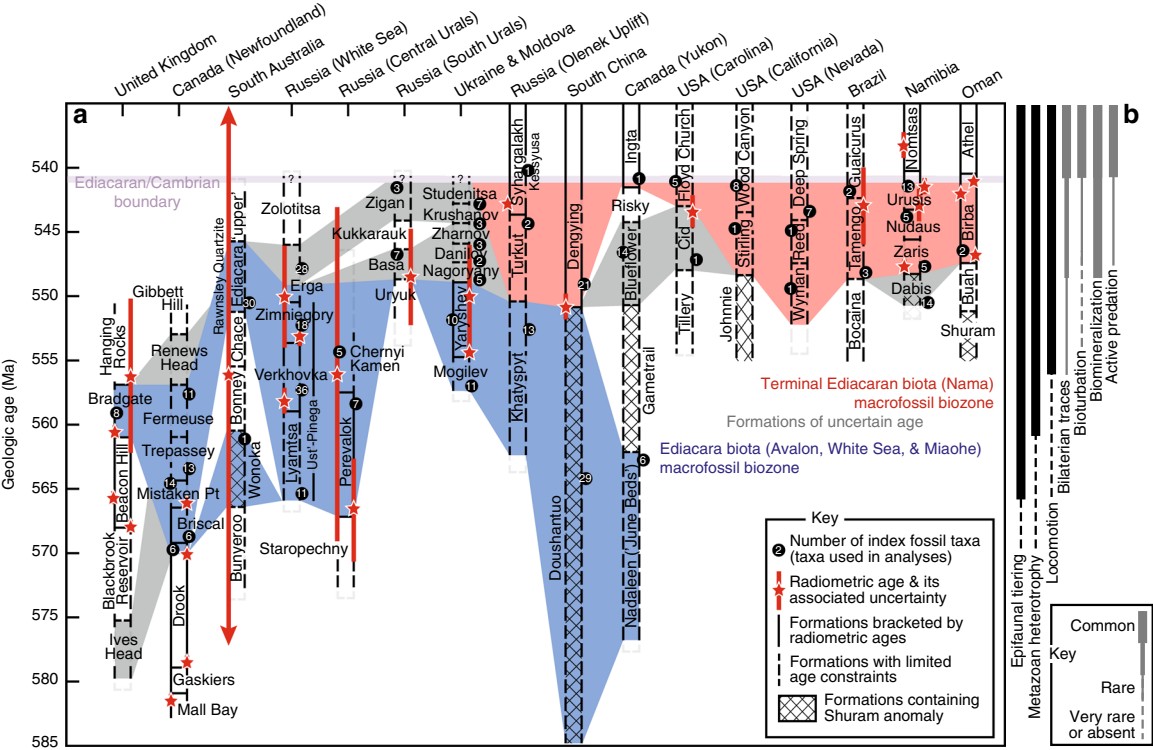

**Fig. 5** Ediacaran stratigraphy and chronology of evolutionary events. **a** Stratigraphic distribution of Ediacaran biozones identified with network analysis. White formations contain no potential index fossils. **b** Ediacaran histories[3] of epifaunal tiering[17], metazoan heterotrophy[14], locomotion[14,69], bilaterian traces[12,55], bioturbation[18,19,70], biomineralization[9,10], and predation[13,20]. Source data are provided as a Source Data file

Fig. 10a), and taxa of the deep biotope are notably missing. Together, the TEB and EBB modules dwarf the small remaining clusters (Fig. 4b). One of these small modules contains the Lantian Formation and the genera of the Lantian Biota[8]. This module shares several genera (i.e., *Doushantuophyton*, *Flabellophyton*, and *Orbisiana*) with the EBB (Fig. 4a). The smallest module, the Ediacaran/Cambrian cluster, overlaps with the TEB, and is comprised of various Siberian formations, where *Cambrotubulus* and *Anabarites* (Cambrian faunal elements) are preserved in rocks of putative Ediacaran age[10,50,51], in some places with *Cloudina*[10,51]. Assortativity coefficients indicate that neither the geographic locations of the formations nor the preservational modes or morphogroups of the taxa represent strong predictors of association (Supplementary Fig. 7).

Units resembling the EBB, TEB, and Ediacaran/Cambrian modules were detected in bipartite networks of Ediacaran formations and taxa, even after genera belonging exclusively to the intermediate and deep modules (Fig. 3a; Supplementary Fig. 10) were removed from the data (Supplementary Fig. 11). The results did not significantly change if ichnogenera were excluded from the data along with the deep-water genera (Supplementary Fig. 12). In all cases, White Sea genera are clustered together in the EBB, and Nama taxa are concentrated in the TEB (Supplementary Figs. 11 and 12).

**Genus richness**. Sample-based rarefaction and extrapolation curves (Fig. 7a, b), which provide estimates of taxonomic diversity vs. sampling intensity, show that the TEB contains significantly fewer genera than the EBB, regardless of sample size (Fig. 4). Unconditional 95% confidence intervals bracketing the curves illustrate that, at the present sampling level, the null hypothesis that the two modules are equal with respect to taxonomic

diversity can be rejected—the observed difference in diversity does not reflect unequal sampling of fossil collection points (Fig. 7a) or geologic formations (Fig. 7b). Extrapolations of genus richness suggest that continued sampling may result in the discovery of additional rare taxa. Nonetheless, the projections predict that, when the modules' sampling levels become high and their taxa accumulation rates are low, the EBB will represent a total diversity two to three times greater than the TEB (Fig. 7a). Estimates of actual genus richness, based on non-parametric (sample-based) richness estimators[52], also suggest that the EBB is two to three times more diverse than the TEB (Supplementary Fig. 14). Analysis of an expanded dataset—one that includes occurrences of simple disc-shaped genera and taxa that may be based on taphomorphs, pseudofossils, or microbial structures—produced identical results (Fig. 7c, d; Supplementary Fig. 15).

The EBB and TEB modules also differ with respect to the taxonomic diversity levels of their White Sea and Nama genera (Fig. 2a). Rarefaction curves demonstrate that, at the present sampling level, the null hypothesis that the two modules are equal with respect to diversity of these taxa can be rejected (Fig. 7e, f). Estimates of actual genus richness suggest that this pattern should hold up with continued sampling of fossils, predicting that the EBB will yield twice as many White Sea and Nama genera as the TEB (Supplementary Fig. 16). With regard to Ediacara-type genera, the analyses also indicate that the EBB is significantly more diverse than the TEB (Fig. 7g, h). Indeed, the EBB may ultimately produce three to four times more Ediacara-type genera than the TEB (Supplementary Fig. 17).

Analyses of Fortunian data (Supplementary Fig. 13) highlight the difference in diversity observed between the EBB and TEB. The EBB and Fortunian are approximately equal with respect to genus richness (Fig. 7a, b; Supplementary Fig. 14), unless the estimates include disc-shaped genera and taxa that may be based

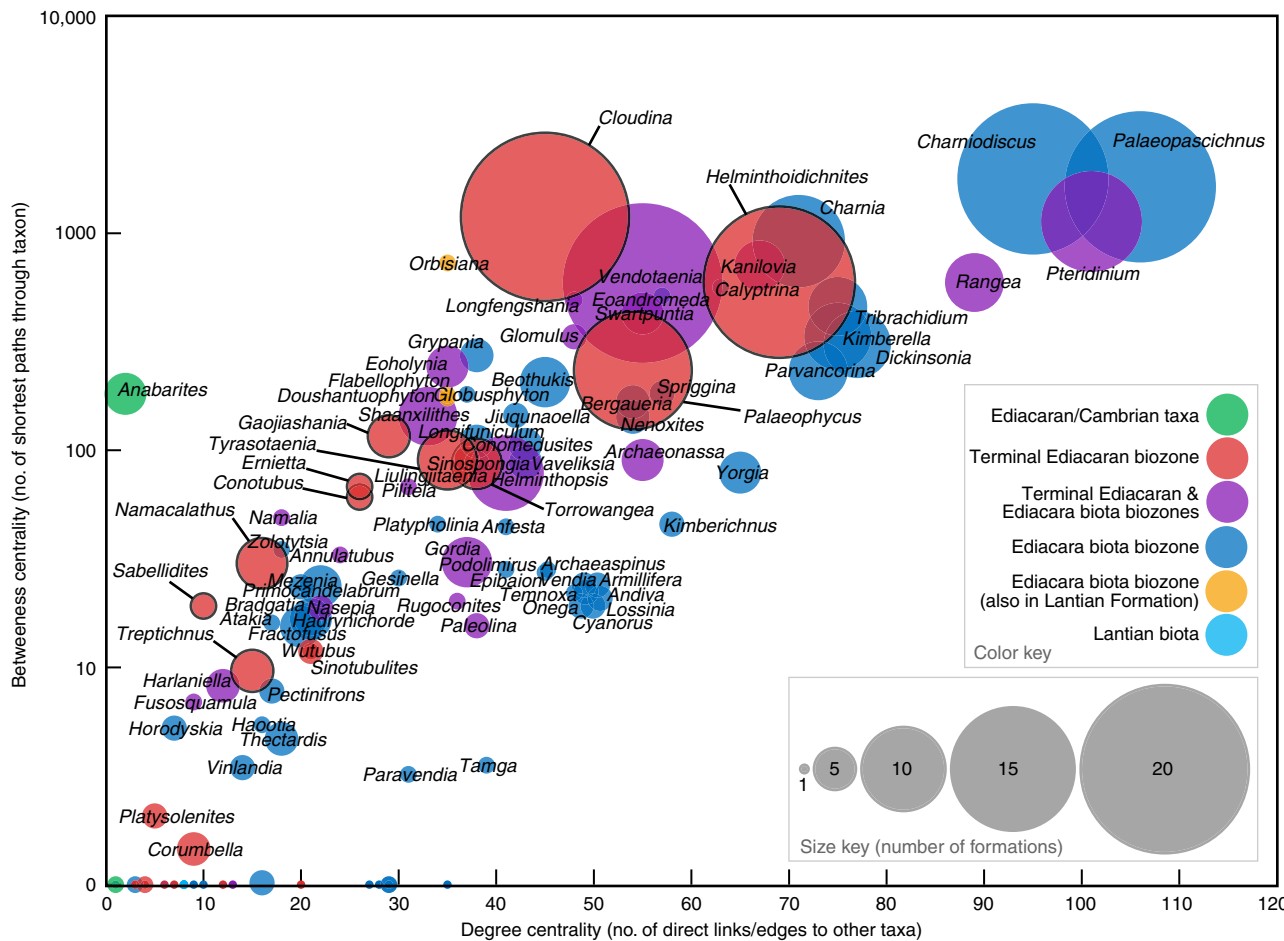

**Fig. 6** Bubble plot showing network centrality scores for stratigraphic index fossils. Each bubble represents an Ediacaran macrofossil genus or ichnogenus in the bipartite network of Ediacaran formations and taxa (Fig. 4). Their colors correspond to biozones in that network (Fig. 4a, b). The diameter of each bubble equals the number of formations that contain fossils of the taxon. The x- and y-axes are node centrality scores. A node's centrality score indicates its relative importance within a network. These scores were calculated from the taxa projection of the bipartite network based on two different definitions of centrality. The x-axis is degree centrality (or simply degree), which is the number of unique links between the taxon and any other. The scores in this plot do not reflect self-loops (i.e., connections between nodes and themselves). Whereas taxa with high degree centrality cooccur with a multitude of others, those with low scores cooccur with relatively few. The y-axis is betweenness centrality, which equals the number of shortest paths that pass through the node in its network. Whereas taxa with low betweenness centrality scores tend to be located at the beginning and end of paths, those with high scores commonly act as bridges for the flow of information. With regard to this plot, the best index fossils are red or blue (i.e., closely associated with one biozone) and have large bubbles and high centrality scores because they are geologically widespread and relatively common. Notable Terminal Ediacaran index fossils have black borders. Source data are provided as a Source Data file

on taphomorphs, pseudofossils, or microbial structures, in which case, the EBB overshadows the Fortunian (Fig. 7c, d; Supplementary Fig. 15). Regardless, the analyses show that the Fortunian is characterized by greater diversity than the TEB, even when differences in sampling intensity are taken into consideration and the data include all body fossils (Fig. 7a–d; Supplementary Figs. 14 and 15).

## Discussion

Various authors have argued that Ediacaran fossils might document two major episodes of extinction and radiation, including ecological reorganization in terminal Ediacaran times and a second event coincident with the Ediacaran–Cambrian transition[1,2,6,23–25,27,28]. According to this hypothesis, Ediacara-type organisms declined in middle-late Ediacaran times[2], leaving behind a depauperate Ediacara biota[23] and ushering in a fauna with recognizable sessile eumetazoans (a 'Wormworld' fauna)[6] that dominated open seas until the onset of the Cambrian

radiation, when this fauna, in turn, disappeared and skeletal animals began their protracted diversification. Three non-exclusive[1] hypotheses have been proposed to explain the disappearance of the Ediacara Biota[2], including (1) a competitive biotic replacement model wherein eumetazoans out-competed and marginalized Ediacara-type organisms;[6,23] (2) a catastrophic, abiotically driven event;[24] and (3) a taphonomic artefact brought about by the loss of environments conducive to Ediacara-type preservation[21]. In any case, attempts to test these hypotheses by measuring global changes in Ediacaran biodiversity have been hampered by issues in biostratigraphy. Some studies have treated the Ediacaran paleocommunities (Figs. 1, 2) as fossil assemblages found in biozones of different ages[1,34]. However, the clusters may alternatively represent ecological-environmental biotopes and/or associations of fossils with shared modes of preservation[31,33,35]. Most efforts to address these concerns have relied on qualitative studies of one or several exemplary regions[33–35] rather than statistical analyses of global data. A few quantitative tests have been conducted[31,32], but they have generally focused on

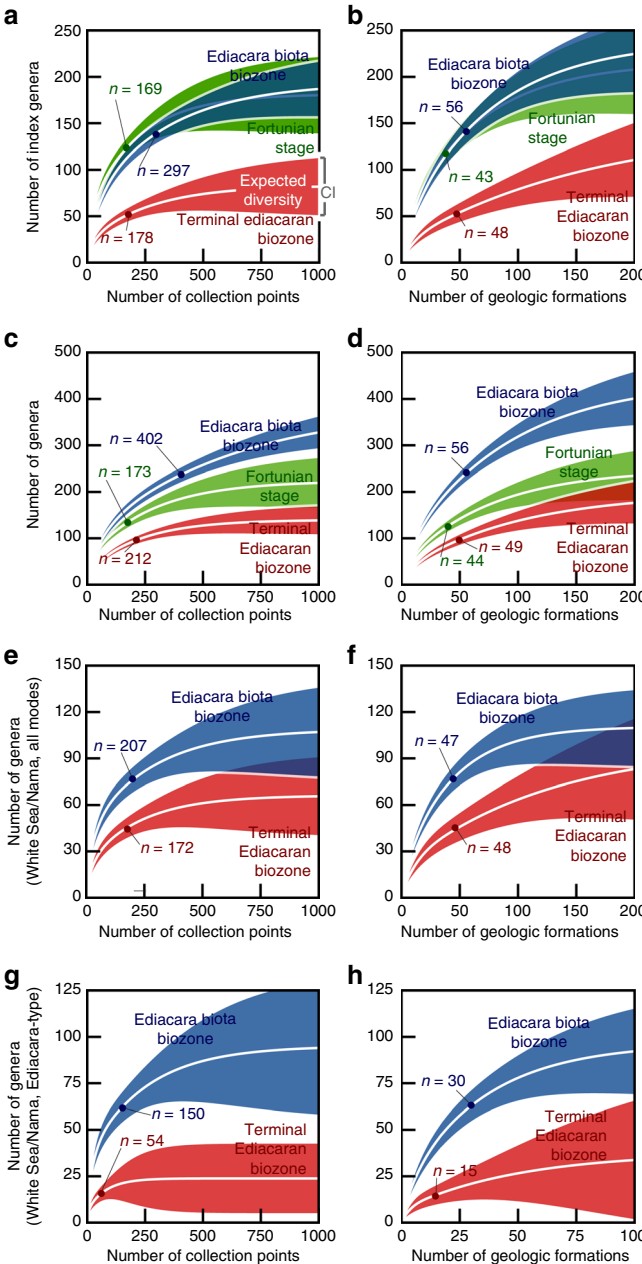

**Fig. 7** Sample-based rarefaction and extrapolation of biozone diversity. The sample-based rarefaction and extrapolation curves show generic richness vs. sampling intensity for various subsets of the data (see Methods). For a given sampling intensity level and biozone, diversity is the number of macrofossil body genera expected when that number of samples is drawn at random without replacement from the biozone. The mean values are bracketed by 95% confidence interval (CI) envelopes[68] (see Methods). Dots indicate current sampling levels, i.e., the total numbers of samples in the dataset. **a, b** Data include only stratigraphic index fossil genera (taxa in Fig. 4; Supplementary Fig. 13). **c, d** Data include all genera, including discs, possible synonyms, and putative taphomorphs and microbial structures. **e, f** Data include only White Sea and Nama genera (Fig. 2a). These paleocommunities occupied similar shelf environments (Fig. 3c), unlike the Miaohe and Avalon clusters. **g, h** Data include only White Sea and Nama genera known from Ediacara-type fossils. **a, c, e, g** Samples are body fossil collection points. Collection points were assigned to biozones based on the community assignments of their formations (Fig. 4; Supplementary Fig. 13). **b, d, f, h** Samples are geologic formations containing body fossils. Source data are provided as a Source Data file

Ediacara-type fossil localities and measurements of their taxonomic similarity[1]. Until now, there has been little integration of fossil occurrence data with metadata on stratigraphy and facies. Our network analysis achieves such data integration by clustering Ediacaran taxa with paleoenvironments and geologic packages, thereby identifying associations with meaningful ecological and stratigraphic boundaries. In this context, our work allows for the formulation of a global biostratigraphic framework, grounded by empirical data and quantitative analysis, for testing the various hypotheses concerning biotic turnover across the Ediacaran–Cambrian transition.

Overall, our networks are based on fossil occurrence data (Table 1). Each link signifies one or more actual cases of specimen sampling and reporting. Consequently, the networks' topographies reflect patterns related to geologic age, preservational pathway, and stratigraphic and geographic location, as all of these variables influence the likelihood of fossil occurrence[36]. The Ediacaran fossil occurrence dataset in this study is larger than its predecessors (Supplementary Fig. 1)[1,22,31,32] and includes comparatively detailed data on occurrences of carbonaceous compressions and skeletal fossils, in addition to the Ediacara-type casts and molds that have generally received most attention. Moreover, the data include trace fossils[53,54], which were omitted from the previous investigations[1,22,31,32], even though they occur with Ediacaran body fossils around the world[3,12,14,18,55].

We explored the networks for community units by applying fourteen algorithms to the data and comparing their results (Supplementary Fig. 5). The COPRA method performed better than other algorithms. It consistently identified the non-overlapping community structures with the highest $Q$ values (Supplementary Fig. 5 and Supplementary Table 6), affirming that it performed best at identifying nonrandom associations of nodes. It also typically returned the fewest communities, making it one of the most conservative approaches for partitioning the dataset. For this reason, we can interpret the modules as macro-level community units, representing the largest and most significant associations of nodes. Although the density of connections within each module is high, the associations are separated by relatively sparse regions of connections[36] that can be interpreted as consequences of biotic turnover of Ediacaran taxa across space and time. For all practical purposes, the unipartite network modules represent paleocommunities, (i.e., associations of taxa that lived and were preserved together at various localities around the world). Along these same lines, the modules consisting of paleoenvironments and taxa constitute biotopes—environments with unique communities of taxa and specific ranges of substrates, hydrodynamic conditions, and light availability. Finally, the modules consisting of formations and taxa signify assemblage biozones, given that they consist of lithologic packages that can be correlated based on associations of taxa.

Altogether, the results provide an empirical framework for exploring how Ediacaran communities were distributed across space and time. Analysis of the unipartite network (Fig. 2a, b) corroborates the results of hierarchical clustering and NMDS (Fig. 1; Supplementary Figs. 2–4), as well as the findings of other studies[31,32], which show that Ediacaran localities and formations can be divided among four clusters based on taxonomic similarity. To a degree, the relative ages of these paleocommunities are unknown, as their sequential appearance through stratigraphy generally varies from region to region (Supplementary Fig. 4). Notably, the multipartite networks do not contain modules that are analogous to these paleocommunities. Assortativity coefficients indicate that the topologies of the multipartite networks do not strongly reflect the preservational modes and/or morphogroups of the taxa or the geographic locations of the geologic units. Therefore, the bipartite modules do not represent artifacts

of taphonomy or paleobiogeography, and instead most likely represent variation in taxa across facies and stratigraphy.

Consistent with previous interpretations[31], network analysis shows that the paleocommunities inhabited environments of varying depth. The Avalon paleocommunity represents a deep-water (slope and basin) biotope[31] that shared only a few taxa with shelf environments (Figs. 2, 3a–c). Shelf environments were characterized by two biotopes that shared many genera (Fig. 3b), with the boundary between them located around wave base. The White Sea and Nama paleocommunities (Fig. 2a) occurred in both biotopes, and occupied similar habitats across shelf settings (Fig. 3c), particularly in shoreface and offshore transition environments where their fossils were commonly preserved (Fig. 3a)[31,33–35,56]. In contrast, the Miaohe paleocommunity primarily lived and was preserved in offshore and slope environments (Fig. 3c)[5,57]. Because few Avalon and Miaohe genera belonged to multiple biotopes, we interpret those associations as environmentally, ecologically, and taphonomically restricted paleocommunities. In conjunction with the bipartite network of Ediacaran formations and taxa (Fig. 4a), the results indicate that the Avalon and Miaohe paleocommunities overlapped in age with the White Sea cluster.

The least ancient (Ediacaran/Cambrian) and most ancient (Lantian) taxa cluster on opposite sides of the bipartite network of Ediacaran formations and taxa (Fig. 4a). *Anabarites* and *Cambrotubulus* are common in the Cambrian[10,51], and although the exact age of the Lantian biota is unclear[8], most correlations suggest that it pre-dates all other Ediacaran macrofossils[29]. Thus, the network exhibits a notable polarity in its structure. Such polarity develops when the stratigraphic position (geologic age) exerts a strong control on network topology[36,44,45]. In this context, we interpret the EBB and TEB modules as assemblage biozones. In general, the TEB assemblage corresponds to the Nama paleocommunity, and the Avalon, White Sea, and Miaohe paleocommunities together constitute the EBB assemblage (Fig. 4a–c). The TEB includes numerous formations located just below the Ediacaran–Cambrian boundary and various units that span the transition (Fig. 5). Where absolute age constraints are available (Source Data), TEB strata are younger than those of the EEB (Fig. 5). The EBB demonstrably occurs below the TEB in South China, Siberia (Olenek Uplift), and Canada (Yukon & Northwest Territories) and below formations of mixed (EBB and TEB) character in Moldova and Ukraine. Additionally, formations of mixed character occur below the TEB in Namibia and the eastern US. Radiometric and other absolute ages suggest that the age of this boundary is approximately 550 Ma (Fig. 5, Source Data).

Centrality scores provide a means of comparing potential index fossils with respect to importance in Ediacaran stratigraphy (Fig. 6, Table 2). Centrality scores indicate that, in addition to the possible coenocytic protist *Palaeopascichnus*[11], the best-connected taxa of the EBB are Ediacara-type fossil genera:

| Table 2 Ediacaran stage-level biozones and their diagnostic taxa | | | | |
| --- | --- | --- | --- | --- |
| **Biozone** | **Carbonaceous compressions** | **Ediacara-type fossils** | **Trace fossils** | **Multiple or other types of fossils** |
| Terminal Ediacaran biozone (551 Ma – 541 Ma) | *Sabellidites* *Tyrasotaenia* | *Ernietta* *Wutubus* | *Helminthoidichnites* *Palaeophycus* *Torrowangea* | *Conotubus* *Gaojiashania* *Cloudina* *Namacalathus* *Sinotubulites* |
| Ediacara biota biozone (571–551 Ma) | *Calyptrina* *Globusphyton* *Jiuqunaoella* *Mezenia* *Sinospongia* | *Andiva* *Archaeaspinus* *Armillifera* *Atakia* *Beothukis* *Bradgatia* *Conomedusites* *Cyanorus* *Dickinsonia* *Fractofusus* *Hadrynichorde* *Kimberella* *Lossinia* *Onega* *Parvancorina* *Platypholinia* *Podolimirus* *Primocandelabrum* *Spriggina* *Temnoxa* *Tribrachidium* *Vaveliksia* *Vendia* *Yorgia* *Zolotytsia* | *Bergaueria* *Epibaion* *Kimberichnus* *Nenoxites* | *Charnia* *Charniodiscus* *Gesinella* *Liulingjitaenia* *Longifuniculum* *Palaeopascichnus* |
| Taxa of uncertain age and/or found in both biozones | *Eoholynia* *Glomulus* *Longfengshania* *Paleolina* *Pilitela* *Vendotaenia* | *Namalia* *Nasepia* *Pteridinium* *Rangea* *Rugoconites* *Swartpuntia* | *Archaeonassa* *Gordia* *Helminthopsis* | *Shaanxilithes* |

Taxa are listed alphabetically by preservational mode. The table includes taxa that have high degree and betweenness centrality scores (both greater than 10) in the bipartite network of Ediacaran formations and taxa (Figs. 4, 6). Source data are provided as a Source Data file.

*Charniodiscus*, *Charnia*, *Tribrachidium*, *Kimberella*, *Dickinsonia*, and *Parvancorina* (Fig. 6). None of the most important elements of the EBB are ichnogenera, and in most cases, the trace fossils of this biozone (e.g., *Kimberichnus* and *Epibaion*) were produced by Ediacara-type organisms[14,58]. In contrast, the most common and best-connected taxa in the TEB are biomineralized and non-biomineralized tubes (i.e., *Cloudina*, *Conotobus*, and *Gaojia-shania*) and metazoan trace fossils (i.e., *Helminthoidichnites*, *Palaeophycus*, and *Torrowangea*), not Ediacara-type fossils (Table 2). Not surprisingly, *Cloudina* represents the best index fossil of terminal Ediacaran strata[9,29]. Our results show that other taxa, which have received attention as potential index fossils of those strata (e.g., *Shaanxilithes* and *Vendotaenia*)[34,59–61], do not by themselves represent good markers.

The network-based biozones illustrate that terminal Ediacaran formations do contain a depauperate Ediacara Biota and unique fauna, represented by skeletons, tubes, and locomotion and feeding traces of bilaterians, in addition to Ediacara-type fossils (Figs. 4–6). Everything being equal, these formations contain significantly fewer genera than those of the older EBB and the younger Fortunian stage (Fig. 7). Thus, our results provide statistically significant evidence of a global, cosmopolitan assemblage unique to terminal Ediacaran strata. This outcome supports the hypothesis that, in late but not latest Ediacaran times (ca. 550–541 Ma)[6,27,28], ecological change involved the initial decline (but not total extinction) of Ediacara-type organisms and the rise of sessile eumetazoans that may have competed with them for space and food[6] (Fig. 5b). The TEB formations also contain traces of various metazoan behaviors, including sediment-mixing[18,19] and macroscopic predation[13,20]. Such behaviors by motile animals may have altered sediment substrates, thereby marginalizing Ediacara-type organisms and reducing their preservational potential[1]. Regardless, the sessile eumetazoans specific to the TEB did not persist into the Cambrian, so network analysis also provides evidence for a pulse of extinction at the Ediacaran–Cambrian boundary. It remains contentious whether, at that transition, emerging cnidarians and bilaterians drove the final ecological collapse of the Ediacara biota, or unlike the latter, they simply survived an environmental perturbation[24].

In summary, we demonstrate the application of network analysis to biostratigraphy and the study of biofacies, and highlight its potential for addressing inter-related paleoecological and stratigraphic problems[36]. Using network analysis, we show that the Ediacaran System contains a number of environmental-ecological biotopes and assemblage biozones. In particular, our work supports a global framework, grounded by empirical data and quantitative analysis, for subdividing the upper Ediacaran System and correlating its terminal strata. Notably, this framework provides a basis for testing hypotheses concerning the evolutionary and ecological changes that led to the Ediacaran–Cambrian transition. Although some variation among Ediacaran paleocommunities reflects environmental and taphonomic gradients, when these effects are accounted for, the data indicate that the evolution of early complex eukaryotes was affected by two biologic turnover events, perhaps the earliest mass extinctions of complex life[23,27]. Whereas the first episode of ecological reorganization led to the establishment of a cosmopolitan metazoan assemblage in the terminal Ediacaran, the second caused the disappearance of the Ediacara biota[2] and the decline of the 'Wormworld'[6], paving the way for the Cambrian radiation of animal life. Future work should aim to determine the durations and drivers of these events. In the meantime, given available information on the chronology of eukaryotic evolution, hypotheses involving biotic replacement and environmental catastrophe merit serious consideration.

## Methods

**Data**. Data on the formations, facies, architectural designs, and nomenclature of trace fossils in the Ediacaran and Cambrian systems were compiled from previously published datasets[53,54], and data on collections, formations, and taxonomy of Cambrian (Fortunian) body fossils were accessed from the Paleobiology Database (PBDB) on February 13, 2018 (https://paleobiodb.org/#/). The original dataset on occurrences of Ediacaran macroscopic body fossils ($n = 1829$) was developed through revision of previously published datasets[1,22,31] and incorporation of additional information from primary and secondary literature sources following search protocols for collection-level sampling set forth by the PBDB (https://paleobiodb.org/data1.1/), wherein a collection represents a set of fossil occurrences co-located at a unique point in geographic and stratigraphic space (Source Data and references therein). We made every effort to exhaust all means of growing the dataset prior to conducting network analysis, and we made no attempts to revise taxonomic work, alter designations of taxa in reports, or assign names to any material of uncertain affinities (i.e., Swartpuntia-like fossils and fossils reminiscent of *Swartpuntia* were not entered as *Swartpuntia*). Occurrences were assigned to 634 stratigraphically and geographically distinct fossil collection points. The dataset includes the geographic location (lat/long), lithostratigraphic unit, country, region (continent), tectonic plate (geoplate in GPlates), and preservational mode of each point along with nearby radiometric ages (Source Data). The paleogeographic locations of the points at 550 Ma were estimated, based on their present coordinates, using the global plate motion model for the Phanerozoic produced by Wright et al.[62] in GPlates. This occurrence data is accompanied by a comprehensive list of Ediacaran and Cambrian macroscopic body fossil genera ($n = 416$) and ichnogenera ($n = 60$). Each taxon was assigned to one of eight taphonomic modes (Supplementary Table 2)—recurrent styles of fossil preservation with unique defining features—and any number of seventeen paleoenvironments (Supplementary Table 4) based on descriptions of fossil facies and preservation in the literature and PBDB. Additionally, each Ediacaran genus was assigned to one of 33 morphogroups[7,48,54] (see Supplementary Discussion), each ichnogenus was assigned to one of 24 trace architectural groups[63] (Supplementary Table 3), and the phyla of the Cambrian animals were compiled from the PBDB. These various groups, in turn, were lumped into more inclusive form and trace categories ($n = 22$) that are more convenient for visualizing data and provide a secondary basis for comparing assortativity coefficients.

**Taxa counted in analyses**. Like other studies[31], every effort was made to include all valid taxa that might serve as index fossils of ecological biotopes and geological biozones in the analyses. Each valid taxon fulfilled the following four criteria. First, the taxon represents a body or behavior of an ancient organism, rather than a microbially induced sedimentary structure (MISS) or any sort of pseudofossil. Second, the taxon is morphologically or architecturally distinct from all others and, in all likelihood, does not represent a junior synonym of another taxon. Taphomorphs, or fossils that have their own taxonomic designations but may represent morphological variants of other taxa[64], were excluded from the data. Third, the taxon has been reported from collection points and formations containing other taxa and, therefore, could be linked to others by network connections. Lastly, the taxon can be recognized and identified with a reasonable amount of confidence. Simple disc-shaped taxa were excluded from the network data. Fossils of these taxa likely represent a mix of discoidal bodies (e.g., medusae), microbial structures and MISSs, and holdfasts of fronds that were not preserved in place. Additionally, a number of the taxa (e.g., *Ediacaria*, *Cyclomedusa*, and *Tirasiana*) may be junior synonyms of *Aspidella*[65], which itself may be the form taxon of a frond holdfast[66]. In any case, most disc-shaped fossils cannot be reliably ascribed to specific taxa. We included data on trace fossils in some of the networks because, unlike genera, ichnogenera generally do not disappear over time, and most trace architectures that originated in the Ediacaran persisted into the Phanerozoic. Therefore, trace fossils bridge the Ediacaran–Cambrian transition and offer information about the evolution of behaviors (mobility and predation) to accompany the record of morphological change.

**Hierarchical clustering**. Hierarchical clustering analyses were performed using functions of the vegan and pvclust packages in RStudio. The taxonomic dissimilarities of Ediacaran geologic formations were calculated using the Jaccard and Kulczynski-2 indices, which are commonly used in paleoecology[31]. To ensure that analyses were based on meaningful measurements of similarity, only formations containing five or more taxa (body fossil genera and trace fossil ichnogenera) were included in the data. Formations were hierarchically clustered into pairs and groups based on their average taxonomic dissimilarities (i.e., the average-linkage method). A multiscale bootstrap resampling with 1000 runs was performed with each analysis using the pvclust function (one-sided statistical test) to calculate approximately unbiased P values for assessing the uncertainty of the results. Clusters with P values larger than 95% are strongly supported by the data (Fig. 1, Supplementary Fig. 2).

**Non-metric multidimensional scaling**. Analyses were performed to corroborate the results of hierarchical clustering[31]. The formations analyzed with hierarchical clustering were ordinated in multidimensional space based on their Kulczynski-2

and Jaccard dissimilarities using the metaMDS function of the vegan package and its default settings in RStudio. The NMDS scores (Supplementary Table 5), ellipsoid hulls, and centroid (mean) confidence intervals were plotted in three dimensions (Supplementary Fig. 3) using the ordirgl and orglellipse functions of the vegan3D package in RStudio. Lack of overlap of two confidence intervals indicates the difference in mean NMDS scores between two clusters is statistically significant.

**Network metrics.** Extended (overlap) modularity scores[47] (Supplementary Tables 6 and 7) were computed for the overlapping and non-overlapping communities in the network projections using the COPRA software[67] written in the JAVA language by S. Gregory (http://gregory.org/research/networks/software/copra.html)[67]. The degree centrality and betweenness centrality scores of nodes (Fig. 6) in the taxa projection of the bipartite network (Fig. 4) were determined using functions of the igraph package and their default settings in RStudio (Source Data). Measures of whole-network properties were also computed using functions of the igraph package for the various networks and network projections in this study (Supplementary Fig. 7 and Supplementary Table 8). Homophily was measured with assortativity coefficients, which are similar to Pearson correlation coefficients, for various continuous and nominal properties of nodes. Assortativity coefficients measuring homophily with respect to degree were determined for all network projections (Supplementary Table 8). Additionally, assortativity coefficients were calculated for taxa projections from data on the preservational modes, morphogroups, and form categories of the genera and ichnogenera. Lastly, assortativity coefficients were calculated for formation projections from data on the G-Plate geoplates, regions (continents), and countries of the geologic units.

**Partitioning networks into non-overlapping modules.** Prior to selecting the COPRA method[67], we partitioned the networks in this study into non-overlapping modules with fourteen community-detection algorithms (see Supplementary Discussion) and then compared the outputs in terms of their numbers of communities and extended modularity scores[47] (Supplementary Fig. 5 and Supplementary Table 6). Weighted and non-weighted versions of the unipartite network were partitioned with the leading eigenvector, Louvain, fast greedy, infomap, walktrap, and edge-betweenness algorithms in the igraph package of RStudio, in addition to the COPRA method of the COPRA software[67]. Non-weighted versions of the bipartite networks were partitioned with the COPRA method; QuanBiMo, LPAwb, and DIRTLPAwb algorithms of the bipartite package in R; LP-BRIM algorithm of the lpbrim package produced by T. Poisot and D. B. Stouffer (http://poisotlab.io/software/) for R; simulated annealing algorithm of the rnetcarto package produced by G. Douclier, R. Guimera, and D.B. Stouffer for R; leading eigenvector and Adaptive BRIM algorithms of the BiMat package in MATLAB; and biSBM algorithm of the C++ code made available by D. Larremore (http://danlarremore.com/bipartiteSBM/). Some of the algorithms do not output a single best fit community structure. The methods lacking output determinism include infomap, walktrap, COPRA, LPAwb, DIRTLPAwb, LPBRIM, Adaptive BRIM, biSBM, and QuanBiMo methods. These methods start from random starting states, and therefore, may produce multiple outputs from a single network. With the exception of the infomap, walktrap, and biSBM algorithms, which did not produce greatly varying results from one run to the next, these methods lacking output determinism were repeatedly applied to each network, and the outputs with the best modularity scores were saved. For each algorithm, the number of runs was determined so the analysis could finish in approximately two hours. The QuanBiMo algorithm was run 100 times; the Adaptive BRIM algorithm was run 1000 times; the LPBRIM algorithm was run 10,000 times; and the COPRA, LPAwb, and DIRTLPAwb algorithms were run 100,000 times. For the COPRA analyses in this comparative work, the $v$ parameter (i.e., the maximum number of communities per vertex) was set to 1, and following the recommendation of the software developer[67], the solutions were extra-simplified throughout the partitioning process to remove communities contained within others.

**Partitioning networks into overlapping modules.** The networks were partitioned into overlapping modules with the COPRA method of the COPRA software[67]. To find the best solutions, we executed COPRA 100,000 times on each network. Again, the solutions were extra-simplified. For this work, we devised and implemented a jackknife resampling and network partitioning procedure to identify the $v$ parameter of each network in this study (Figs. 2a, 3a, 4a; Supplementary Figs. 10a, 11a, and 12a). For each network, a single node was removed from the data, the network was partitioned using the COPRA method ($v = 1$), and the number of non-overlapping, non-singleton communities ($n$) was recorded. Then, the node was reinserted into the network, and the steps were systematically repeated, so that every node in the network was omitted once and a distribution of $n$ values was produced (Supplementary Fig. 6), where the total number of $n$ values equals the size of the network (i.e., the number of nodes). Following this procedure, the $v$ parameter is equal to the maximum $n$ value in the distribution.

**Sensitivity analysis.** To assess the sensitivity of the network partitioning results to the level of sampling, we analyzed the effects of omitting links and nodes from the

networks (Supplementary Fig. 8). In this analysis, links connect taxa to collections, paleoenvironments, and formations. For each network, links were randomly subsampled from the data in order to identify a subnetwork. Next, the number of nodes omitted from the subnetwork as a consequence of the subsampling procedure was determined. Then, the COPRA algorithm ($v = 1$) was applied 100,000 times to the subnetwork, and the community structure with the highest modularity score was identified. Finally, the best subnetwork partition was compared to the best network partition (i.e. the reported community structure). This final step involved calculating a normalized mutual information (NMI) score with igraph package function in RStudio. In network analysis, NMI is a common measure of similarity (linear and nonlinear dependence) for two clusterings[49]. These scores are similar to Pearson correlation coefficients with values between 0 (no dependence) and 1 (identical clusterings). Our NMI calculations assume that each omitted node represents its own module in a subnetwork. Overall, these steps were repeated one hundred times at various sampling levels, each corresponding to a percentage of links. Using this procedure, we compiled distributions of NMI scores and omitted node counts vs. sampling level. High NMI scores, particularly those paired with high omitted node counts, indicate results robust to variation in the data. To produce a null model for testing the statistical significance of the NMI scores, we repeated the procedure, except each NMI score was calculated for a pair of networks that were randomly produced with properties (size and degree distribution) based on the network and subnetwork. Unipartite and bipartite null models were generated in RStudio using functions of the igraph (sample_degseq) and bipartite (vaznull) packages, respectively. In this case, the null hypothesis is that the network and its subnetworks do not have comparable community structures at a given sampling level (i.e., the observed NMI scores reflect random similarities). If the majority (95%) of the observed NMI scores are greater than those of the random networks (one-sided statistical test), then the null hypothesis can be rejected.

**Randomization testing.** A number of methods have been proposed for determining whether a community structure is statistically significant or, conversely, if it could have arisen due to chance[49]. Typically, a high modularity score is a good indicator of community structure[46], but not all networks with high modularity have strong community structure. To assess if the community structures reported in this study arose due to chance (Figs. 2a, 3a, 4a; Supplementary Figs. 10a, 11a, and 12a), we performed a randomization test (Supplementary Fig. 9). For a unipartite network, the null hypothesis of this test is that its observed modularity score equals the value of a random network of matching size and degree distribution, i.e. a network that has the same numbers of nodes with various degrees (numbers of connections). The null hypothesis is essentially the same for bipartite networks. However, each projection in a bipartite network has its own modularity score, so the null hypothesis states that one or both scores are equal to those of random networks. To test this hypothesis for each network, the links among nodes were randomized using functions of the igraph and BiRewire packages in RStudio. Nonetheless, the nodes' degree distribution was preserved for all projections in the random network. The randomized network was then partitioned using the COPRA method and $v$ parameter that was applied to the original network, and the modularity of the community structure was recorded. These steps were repeated 100 times for each network, producing one distribution of modularity scores per unipartite network and two distributions of modularity scores per bipartite network (one for each projection). These distributions were used to calculate $P$-values and $Z$ scores to test the null hypothesis (i.e., a one-sided statistical test). The $P$-value is the probability of discovering a community structure with a higher modularity score if the connections among nodes were randomly distributed (i.e., there is no meaningful community structure). If the $P$-value of a unipartite network or both $P$-values of a bipartite network are less than alpha ($\alpha$) at the 90% (0.10), 95% (0.05), and/or 99% (0.01) confidence levels, the null hypothesis can be rejected, and its community structure is considered statistically significant. A $Z$ score greater than 1 also suggests that an observed community structure is significant[49].

**Network visualization.** Networks were visualized in RStudio using functions in the following packages: igraph, GGally, ggplot2, ggnetwork, and intergraph. Static network graphs (Figs. 2a, 3a, 4a; Supplementary Figs. 10a, 11a, and 12a) were generated using the ggnet2 function of ggplot2 and its default parameters, and nodes of equal size were placed without self-loops according to the Fruchterman-Reingold force-directed algorithm.

**Generic richness data.** The taxonomic diversities of the biozones in this study were estimated from sample-based incidence (i.e., presence/absence) data with rarefaction, extrapolation, and non-parametric richness estimators (Fig. 7; Supplementary Figs. 14–17). In this work, samples are fossiliferous formations and fossil collection points, which vary in number among the biozones. Biozone assignments of Ediacaran formations were taken directly from network analysis results (Fig. 4). Collection points, on the other hand, were assigned to biozones based on their formations. Four subsets of samples were analyzed. The first subset is comprised of all samples of body fossils. Genera in this subset include simple discs (e.g., *Aspidella*) and possible junior synonyms as well as taxa that may be based on taphomorphs, pseudofossils, and microbial structures. The second subset

consists of all samples of index fossil genera (i.e., the morphologically distinct taxa that define the biozones, Fig. 4; Supplementary Fig. 13). In contrast, the third and fourth subsets exclude all samples associated with the deep biotope (i.e., the Bradgate, Briscal, Drook, Fermeuse, Mistaken Point, Nadaleen, and Trepassey samples) as well as all genera assigned to the environmentally restricted Avalon and Miaohe paleocommunities (Figs. 2a, 3a–c). Whereas the third subset consists of all remaining samples of White Sea and Nama taxa, the final subset is comprised of samples of taxa that are known from Ediacara-type fossils and assigned to those paleocommunities (Fig. 2). Ichnogenera occurrences were not included in any of the subsets.

**Sample-based rarefaction and extrapolation.** A number of sample-based rarefaction analyses were performed using the EstimateS software[68] to compare the three (Ediacara biota, Terminal Ediacaran, and Fortunian) stage-level biozones (Supplementary Fig. 13) in terms of taxonomic diversity (genus richness) estimates and sampling intensity (Fig. 7). For sampling intensity 1:$n$ (where $n$ equals the number of samples within each biozone), the expected number of taxa and unconditional 95% confidence interval was calculated for 1000 runs using established analytical methods[68], which duplicate the results of conventional subsampling techniques (Source Data). Additionally, non-parametric methods for extrapolation[68] were used to estimate the expected numbers of taxa that would be found in augmented samplings with greater numbers of samples. These exact analytical methods were also used to calculate unconditional 95% confidence intervals for the extrapolated values. The unconditional 95% confidence intervals in this rarefaction and extrapolation work can be used for hypothesis testing. The null hypothesis is that two assemblages (i.e., biozones) are equal with respect to their taxonomic diversity. If the confidence intervals of two biozones do not overlap at the current sampling level of the biozone with fewer samples, the null hypothesis can be rejected, and the observed difference in generic diversity is considered statistically significant. Values extrapolated beyond the current sampling level may provide evidence to the contrary, particularly if the rate of taxa accumulation is significantly greater in one assemblage than the other. The amount of variance, however, generally increases with the level of extrapolation, so interpretations of the data should holistically consider the shapes of the rarefaction/extrapolation curves as well as their uncertainties at various sampling levels.

**Estimation of taxonomic richness.** Five common non-parametric richness estimators were used to estimate the generic diversities of the three stage-level biozones (Supplementary Fig. 13) as functions of sampling intensity (Source Data). These analyses were conducted using the EstimateS software[68] (Supplementary Figs. 14–17). The estimators correct richness values observed in incidence data by adding terms based on the frequencies of rare taxa (i.e., taxa represented in only one sample or a few)[52]. They include the Chao-2 (classic formula), bootstrap, first-order jackknife, and second-order jackknife estimators as well as the incidence coverage-based estimator (ICE). For sampling intensity 1:$n$ (where $n$ equals the number of samples within each biozone), samples were randomly selected without replacement from each biozone, and the number of genera was determined using each estimation method. This subsampling was repeated 1000 times for each biozone, and the mean number of genera was calculated for each sampling intensity level. The distributions of iterated bootstrap, second-order jackknife, and ICE mean values were used to calculate conditional variance values and 95% confidence intervals for these estimators. Conversely, exact analytical methods[68] were used to derive unconditional variance values and 95% confidence intervals for the mean second-order jackknife and Chao-2 estimators. Unlike the conditional confidence intervals, the unconditional intervals do not converge to zero at the maximum sampling intensity level. If the unconditional 95% CIs of two assemblages do not overlap for a given estimator at this level, the data indicate that the two assemblages are statistical different[68]. On the other hand, if two conditional CIs do not overlap, the results simply suggest that the smaller reference sample was not drawn from the larger one.

**Code availability.** The authors declare that the study does not include results produced using custom software or mathematical algorithms. All codes are available from the corresponding author upon reasonable request.

**Reporting summary.** Further information on experimental design is available in the Nature Research Reporting Summary linked to this article.

## Data availability
The authors declare that the main data supporting the findings of this study are available within the article and its Supplementary Information files. The source data underlying all the figures and tables are provided as a Source Data file.

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

## Acknowledgements

This work was supported by a Keck Foundation project (The Co-Evolution of the Geo- and Biospheres: An Integrated Program for Data-Driven Abductive Discovery in Earth Sciences). Additional support was provided by the Deep Carbon Observatory, the Alfred P. Sloan Foundation, a private foundation, the Carnegie Institution for Science, NSERC, and the Russian Science Foundation (No. 17-17-01241).

## Author contributions

A.D.M., M.B.M., J.D.S., P.F., R.M.H. and A.H.K. designed the research; A.D.M., N.B., T.H.B., L.A.B., and M.G.M. collected the data; A.D.M., A.E., A.P. and F.P. analyzed the data. A.D.M., M.B.M., and A.H.K. wrote the paper with significant input from all of the authors.

## Additional information

**Competing interests:** The authors declare no competing interests.

