## [Peer Review File · Nature Communications]

Reviewers' Comments:

Reviewer #1:

Remarks to the Author:

Review of NCOMMS-18-29230 – Dr Jennifer Hoyal Cuthill

This paper addresses the important question of Ediacaran-Cambrian diversity and community structure and applies some interesting and methodologically robust new approaches in extensive quantitative analyses, for which the authors should be commended. However, I think the write-up would benefit from some extra work to bring out the novelty in the results.

Which are the new results or conclusions that these new methods bring, which hadn't already been shown using other qualitative and quantitative approaches? E.g. The importance of different facies (e.g. communities from different water depths) has been previously discussed e.g. Grazhdankin 2004 as have late/terminal Ediacaran mass extinctions (including in previous papers by the authors). Can you then reorganise your results and discussion so that these novel points are most prominent. Other parts confirming previous findings could be made more concise and some moved to the supplementary material if needed to give you space to focus on new findings.

The authors have gone to the trouble of assembling their own datasets of fossil occurrences. However the paper also treats the results rather literally and the main text doesn't discuss the effect of any potential biases affecting the compilation of the underlying data itself. Therefore I think it would be helpful to provide some sensitivity analyses and additional discussion e.g. to tell us how many links would have to be added to the underlying data (or removed) before your main conclusions broke down.

I've had a look at the underlying data and it appears there are some relevant occurrences of body fossils which are not included in the study. For example *Bradgatia* is found in relatively young Ediacaran localities in Australia but only appears to be listed in the dataset for older localities in Canada and the UK (e.g. examples are figured and mentioned, though not with a strongly stated taxonomic affiliation, in Gehling and Droser 2013, they are also accessioned under *Bradgatia* in the South Australian Museum). Specimens described as *Swartpuntia*-like are also known from early Cambrian Australia (Jensen et al 1998). These are just a couple of examples that spring to mind and I haven't thoroughly checked all taxa in the dataset. They do illustrate though that the authors' results will be dependent on whatever search strategy was used to compile the data and other, more inclusive criteria could have been used. This search protocol should be described fully in the methods (at the moment it isn't) and mentioned briefly in the main text as well.

If the authors perform a quantitative sensitivity analysis as suggested above this would allow the reader to form an impression of how robust the conclusions are (e.g. regarding the distinct clusters suggested) e.g. to links added by a more inclusive data gathering procedure or from new fossil discoveries (which are ongoing for this time period). Can you even state your results probabilistically?

Can you also relate your suggested mass extinction events to standard expectations of background versus mass extinction rates? Your title talks about pulsed extinctions but it seemed there could be much more evaluation of this idea in the results and discussion. I would suggest that a little more secondary analysis of the data you generated might really differentiate your study from previous (especially qualitative) work.

Overall I found the paper to be well written but on reading it there appeared to be an unjustified jump from the results which mentioned turnover in geographic space or time as possible explanations for

the distinct clusters identified and the conclusions which only mention time. Is this justifiable? If so, please make this justification clearer. If not, then the discussion, abstract and title should be modified accordingly.

The authors suggest that their quantitative data could form the basis for new geological stage divisions and presumably corresponding diagnostic taxa. If so, you should actually add a Table showing the break times and diagnostic taxa recovered by your analysis. I believe this would be a concrete new suggestion.

I also attach an annotated pdf highlighting a few typos and additional minor suggestions.

Reviewer #2:

Remarks to the Author:

This study takes full advantage of the possibilities of network analysis in assessing biozones among Ediacaran faunas and occurrences. The methods are state-of-the-art and the results are presented and discussed in a clear way. I think the results and also the methods are relevant for a large audience and I can recommend the publication of this manuscript largely as it is, with only two minor comments.

Lines 441-442 "was affected by two biologic turnover events, perhaps the earliest Phanerozoic-style mass extinctions." Are there also "Precambrian-style mass extinctions"? I don't find this term "Phanerozoic-style" very useful here – it could simply be omitted.

Line 525-527 the measured network properties include, among others, diameter, transitivity, and edge-density. But I've not seen these three properties used or discussed anywhere in the manuscript (I didn't check the suppl. text, though); perhaps these could just be omitted from the methods description?

Reviewers' comments:

Reviewer #1 (Remarks to the Author):

Review of NCOMMS-18-29230 – Dr Jennifer Hoyal Cuthill

This paper addresses the important question of Ediacaran-Cambrian diversity and community structure and applies some interesting and methodologically robust new approaches in extensive quantitative analyses, for which the authors should be commended. However, I think the write-up would benefit from some extra work to bring out the novelty in the results.

Which are the new results or conclusions that these new methods bring, which hadn't already been shown using other qualitative and quantitative approaches? E.g. The importance of different facies (e.g. communities from different water depths) has been previously discussed e.g. Grazhdankin 2004 as have late/terminal Ediacaran mass extinctions (including in previous papers by the authors). Can you then reorganise your results and discussion so that these novel points are most prominent. Other parts confirming previous findings could be made more concise and some moved to the supplementary material if needed to give you space to focus on new findings.

>>> We have revised the manuscript to address these comments, adding a new paragraph at the start of the Discussion section as well as additional text toward the end of the section.

>>>It is true that some authors have hypothesized that the Ediacaran communities represent biotopes, while others have argued that they may represent biozones, endemic 'faunas', or taphonomic windows. However, for a number of reasons, previous works have failed to address the issue and the Ediacaran clusters remain an ongoing topic of debate. Grazhdankin (2004), for example, based his conclusions on observations of Ediacara-type fossils in the White Sea and Ural Mountains regions of Russia. Gehling and Droser (2013, Geology) made similar arguments based on observations of fossils in South Australia. Besides representing regional investigations of narrow focus, these works do NOT include quantitative analyses, so it is unclear if their conclusions are statistically significant. Some quantitative studies have been conducted (Boag et al., 2016; Waggoner, 2003), but these studies focus on Ediacara-type fossil localities and their taxonomic similarities. There has been no integration of data on fossil occurrences with metadata on stratigraphy and facies. Such integration is necessary to address the issue.

>>> Our research represents a significant advancement over prior works, as it thoroughly and quantitatively explores the issue using a new method/approach and wholesale dataset. The dataset represents a marked improvement upon its predecessors (Boag et al., 2016; Waggoner, 2003), as it is far bigger and possesses better representation of non-Ediacara-type fossils (e.g. skeletal fossils and carbonaceous compression). Our work utilizes global data, rather than data on fossils of

a specific type, region, or outcrop, and our method/approach to 'clustering' is unique, in that, it is not based solely on taxonomic similarity like previous papers. This method integrates data on fossil occurrences with metadata on fossil stratigraphy and facies.

>>> Similar issues apply to prior work on the late/terminal Ediacaran extinction(s). Smith et al. (2016; 2017) argued for an extinction at the Ediacaran-Cambrian boundary based on a few sections in the Great Basin (SW USA). BUT again, these works are narrowly focused AND they provide no quantitative analysis, such as specimen- or sample-normalized estimates of diversity on a global scale. For these reasons, the 'extinctions' in these sections could reflect local phenomena affecting habitats or fossil preservation. Darroch et al. (2015, Proc. Royal. Soc. B) were also focused on a specific area. They argued for the existence of a 'mass extinction' in the Ediacaran based on specimen-rarefied measures of diversity at Swartpunt, a locality in Namibia, which is considerably less diverse than other (older) Ediacaran sites. HOWEVER, again, the low diversity at Swartpunt may reflect local phenomena.

>>> In this context, our work is novel for a number of reasons. (1) We provide a new global biostratigraphic framework, which is grounded in quantitative analysis. (2) We use this framework to test the extinction hypothesis, and provide sample-normalized estimates of diversity on a global (NOT local) scale. (3) Our quantitative analysis provides the first statistically significant evidence for a global, cosmopolitan assemblage unique to terminal Ediacaran strata, thereby providing support for the hypothesis that there were two pulses of extinction during the Ediacaran-Cambrian transition.

>>> Anyway, we hope our revisions to the Discussion highlight the novelty of our work.

(Smith et al., 2016; Smith et al., 2017)

(Darroch et al., 2015)

(Grazhdankin, 2004)

(Gehling and Droser, 2013)

The authors have gone to the trouble of assembling their own datasets of fossil occurrences. However the paper also treats the results rather literally and the main text doesn't discuss the effect of any potential biases affecting the compilation of the underlying data itself. Therefore I think it would be helpful to provide some sensitivity analyses and additional discussion e.g. to tell us how many links would have to be added to the underlying data (or removed) before your main conclusions broke down.

>>> We pulled data on fossil collection points from primary and secondary literature sources, which are listed in the source data. This approach, which has become popular in paleontology in recent years with the rise of the Paleobiology Database, is fairly standard. Suffice it to say, we made every effort to limit any conscious biases when compiling the data, and followed every available thread to find additional data prior to running the analyses and preparing the manuscript. Of course, our efforts do not exclude the possibility that biases may exist.

>>> To address this concern, we performed a sensitivity analysis and incorporated the results into the manuscript. The analysis is described in detail in the methods. Overall, the analysis assesses how the results (the modules of the networks) vary with the amount of data (numbers of links and nodes).

>> In short, we randomly sampled the links of each network at various levels (taking 99%, 95%, 90%, 75%, 50%, and 25% of the links), generating 100 random sub-networks at each level. To each randomly generated sub-network, we applied the COPRA algorithm (as we did to the total dataset) and identified its community structure. We then compared each randomly generated sub-network to its network, calculating a similarity index called normalized mutual information (NMI). This index is similar to a Pearson correlation coefficient (min: 0; max: 1). When networks and subnetworks have similar community structures, their NMI scores are high; when they are dissimilar, their values are low. Finally, for each level of sampling, we determined the distribution of NMI scores, and compared it to a null model.

>>> These sensitivity analyses show that the results (community structures) do not substantially vary, even when, at least, ten percent of the links and five weakly connected taxa are removed from each network. In other words, the results are insensitive to minor variations in the data.

>>> When it comes to community detection algorithms, it is very difficult to compare one result to another. In this work, there are three basic variables: (1) the number of modules, (2) the average number of nodes per module, and (3) the compositions of the modules (i.e. their combinations of nodes). The first variable provides little value, as the nodes may be spread out evenly among all modules or concentrated in just a few. In the latter case, two clusterings may differ with regard to the number of modules even though they closely resemble each other with respect to the other variables. Likewise, the average number of nodes can be misleading—this value may be the same for two clusterings, even though their modules contain different combinations of nodes. For these reasons, the third variable (the relative assignments of nodes) tends to be the most meaningful, even though it is the most difficult to measure. In any case, we focus on normalized mutual information (NMI) scores because they reflect all three variables.

>>> We generally found that networks with NMI scores >0.4 were similar to each other, and those with scores >0.7 were virtually identical, except for missing taxa.

>>> We regard this sensitivity analysis as a highly conservative test of robustness. All of the links had an equal probability of being sampled, even though they are not equally supported by data. Whereas some links represent many specimens and reports, others represent just one. Hundreds of specimens of *Dickinsonia* have been reported from Rawnsley Quartzite. In contrast, the Dabis Formation has produced the-one-and-only specimen of *Ausia*. Yet, for this analysis, the *Dickinsonia*-Rawnsley_Quartzite and *Ausia*-Dabis links are equally weighted, even though the *Ausia*-Dabis link should be removed more often, as it backed by less empirical data and typifies a link that is sensitive to sampling bias. Importantly, central nodes like *Dickinsonia* tend to represent

more data than peripheral nodes like *Ausia*. In addition, central nodes exert greater influence on network structure than peripheral nodes, as they have more connections to other nodes. As a result, the removal of a link to a central node will result in a greater change in community structure than the removal of a link to a peripheral node. Thus, it is likely that the sampling procedure over-estimates the effects of random sampling.

>>> For our purposes, a conservative test is acceptable, if not desirable, as it limits the potential for Type II errors. So, there is no problem. That said, one should consider the conservative nature of the test when assessing its results.

>>> To improve upon the accuracy of the test, we would need additional data, allowing us to better account for real differences in sampling effort. The legacy data does not include information on specimens or reports, and we did not develop the new dataset for this purpose. It should be addressed in future research.

I've had a look at the underlying data and it appears there are some relevant occurrences of body fossils which are not included in the study. For example *Bradgatia* is found in relatively young Ediacaran localities in Australia but only appears to be listed in the dataset for older localities in Canada and the UK (e.g. examples are figured and mentioned, though not with a strongly stated taxonomic affiliation, in Gehling and Droser 2013, they are also accessioned under *Bradgatia* in the South Australian Museum). Specimens described as *Swartpuntia*-like are also known from early Cambrian Australia (Jensen et al 1998). These are just a couple of examples that spring to mind and I haven't thoroughly checked all taxa in the dataset. They do illustrate though that the authors' results will be dependent on whatever search strategy was used to compile the data and other, more inclusive criteria could have been used. This search protocol should be described fully in the methods (at the moment it isn't) and mentioned briefly in the main text as well.

>>> We welcome comments like this one because they may ultimately help us and others to grow the dataset for future work. It is fair to say that Ediacaran workers (and paleontologists in general) often disagree about taxonomy. It is also fair to say that no sampling of data is ever complete or perfect. Nonetheless, our search protocol represents a fairly standard approach to compiling data in paleontology, and we are confident that the data is adequate for our purposes, as it is a substantial improvement on the source data of previous studies of Ediacaran fossils (Boag et al., 2016; Shen et al., 2008; Waggoner, 1999; Waggoner, 2003). Indeed, the new sensitivity analysis shows that the results are insensitive to minor variations in the data.

>>> We have revised and added text to the Methods section, which clarifies our search protocol. It is allied with the standard search protocol of the PBDB, so more information can be found on the PBDB website (<https://paleobiodb.org/data1.1/>). This approach has become standard in the field.

>>> Our strategy was simple—sample primary and secondary literature for data on fossil collection points (specifically, occurrences of genera). We used conventional

resources to guide us in compiling data, like search engines as well as previously published datasets and the literature itself.

>>> This comment identifies two key aspects of our search protocol, which merit some explanation here. First, all of our data are based on published reports of fossils and fossil taxa – we did not incorporate data on specimens held in museums or other collections, unless of course, they were described in publications. Second, we made no effort at taxonomic revision. We did NOT revise previous taxonomic work, assign new names to any specimens, or guess the assignments of fossils of uncertain affinities.

>> In some cases, we disagreed with published taxonomic assignments of fossils described in the literature. Nonetheless, we made every effort to keep the data consistent with published opinions and treatments. In this way, the work is reproducible.

>>> With regard to the reviewer's comment, we did not include data on fossils of uncertain taxonomic position(s). This means, 'Swartpuntia-like' fossils were not counted as *Swartpuntia*. Although those fossils may resemble the aforementioned genera, they may represent those specific taxa, or alternatively, represent allied (but distinct) genera.

>>> We acknowledge that the reviewer has seen rangeomorphs labeled as 'Bradgatia' in the South Australia Museum. BUT to our knowledge, those specimens have not been described or illustrated. Moreover, it seems that Gehling and Droser (2013) regard them as rangeomorphs, similar but not necessarily referable to *Bradgatia*: *'the Mass-Flow Sand facies includes taxa best known from the Nama Group in southern Namibia such as Nasepia and Archaeichnium, as well as rangeomorphs more reminiscent of forms such as Bradgatia from the Mistaken Point assemblage in Newfoundland and Leicestershire.'* This may be the reason that the authors listed *Nasepia* and *Archaeichnium*, but not *Bradgatia*, among the genera in the Rawnsley Quartzite at Nilpena (Table 1, Gehling and Droser, 2003, Geology). There are many types of rangeomorphs, and it is not outside the realm of possibility that they discovered a new genus. With the uncertainty, we did not include this fossil occurrence in the dataset.

>>> It is highly unlikely the inclusion of this possible occurrence of *Bradgatia* in the dataset would have changed the results. The Nilpena fossils are largely found in offshore shelf and offshore shelf transition facies. In Figure 3, *Bradgatia* already belongs to the deep and intermediate biotopes. In Figure 4, *Bradgatia* and the Rawnsley Quartzite belong to the same biozone. So, it is unlikely that some additional links to *Bradgatia* would drastically alter the results or conclusions of our network analysis.

>>> We are aware of the 'Swartpuntia-like' fossils reported from the lower Cambrian of South Australia. Other 'Swartpuntia-like' fossils (Hagadorn et al., 2000) have been reported from lower Cambrian in the Great Basin (western US). Some among us are doubtful that any of these fossils are actually *Swartpuntia* – they are similar to *Swartpuntia* but may represent one or more different genera. In any case, the fossils are not actually called *Swartpuntia*, so we did not include them in the data. Additionally, the

lower Cambrian data in our study comes from the Paleobiology Database – for simplicity, we did not revise or expand the PBDB data.

>>> We acknowledge that there are other fossils, like these *Swartpuntia*-like and *Bradgatia*-like forms, which could have been included in the data. In many of these cases, the fossils are poorly preserved, and cannot be identified with confidence. So, our protocol serves to limit the effects of possible taxonomic errors.

If the authors perform a quantitative sensitivity analysis as suggested above this would allow the reader to form an impression of how robust the conclusions are (e.g. regarding the distinct clusters suggested) e.g. to links added by a more inclusive data gathering procedure or from new fossil discoveries (which are ongoing for this time period). Can you even state your results probabilistically?

>>> See our comments above regarding the new sensitivity analysis. This analysis shows that the results are robust to minor variations in the data.

>>> The answer to the reviewer's question is, yes. We can (and do) state the results probabilistically. The manuscript now includes a number of statistical tests, including tests of network modularity, rarefaction results, and taxonomic richness estimates, in addition to the sensitivity analysis. Overall, our analyses identify patterns in the data (e.g. modules) and demonstrate that those patterns are probabilistically robust (i.e. did not arise due to chance).

>>> For example, we use a probability test involving data randomization to assess whether or not our networks are actually modular, or if alternatively, their modules arose as a result of chance (given the sizes of the networks). In this case, for each network, the null hypothesis is that its modularity score does not significantly differ from the score a random network of corresponding size and degree distribution. Based on our analysis, we can reject the null hypothesis for each network – it is highly unlikely that any of the modules arose due to chance. Thus, we do state some results probabilistically.

>>> The sensitivity and randomization analyses show that the modules/units, which we report, are robust. What do the modules/units represent? The answer to this question is open to interpretation. We argue for the simplest interpretation of the modules, that is, that they represent biozones and biotopes, as they variably contain mixtures of taxa, environments, and geologic formations.

>>> It's hard to imagine a better data gathering procedure. Ediacaran fossil localities are numerous BUT their number is hardly beyond comprehension at this moment in time. In addition, the best sites are very well known and intensely studied, given their importance for understanding the origin of complex eukaryotic life and animals. We can be fairly certain that localities, which we did not sample, are not very productive.

>>> The community structures of our networks are strongly influenced by the most common and widespread taxa (e.g. *Cloudina*, *Charniodiscus*, etc.). Although one can

expect new fossil discoveries in the future (new localities, new formations, new taxa), it is unlikely that these discoveries will profoundly affect the data pertaining to these major taxa. Indeed, new discoveries will probably be rare and poorly connected taxa (i.e. taxa that haven't been reported yet). Such taxa have little influence on network structure, as demonstrated by the sensitivity analysis. In this context, it is doubtful that new discoveries will drastically alter the results.

Can you also relate your suggested mass extinction events to standard expectations of background versus mass extinction rates? Your title talks about pulsed extinctions but it seemed there could be much more evaluation of this idea in the results and discussion. I would suggest that a little more secondary analysis of the data you generated might really differentiate your study from previous (especially qualitative) work.

>>> Regretfully, no. For the various reasons below, any discussion of this topic would be very speculative, and we would prefer to keep our discussion grounded by data.

>>> While we can confidently state that the Ediacaran-Cambrian transition contains two distinct intervals of taxonomic and ecologic turnover (the 'extinctions'), we do not yet have enough data to calculate true extinction 'rates' and to compare them to those of other ages of Earth history.

>>> The conventional method is to calculate extinction rates for geologic intervals (e.g. stages) based on appearances and disappearances of taxa around their upper and lower boundaries (Alroy, 2010; Alroy, 2015; Foote, 2000). In particular, most rate-estimates are based on occurrences of 'three-timers,' or taxa known from three consecutive stages

>>> We do not yet have the stratigraphic resolution (or sufficient data) to calculate meaningful extinction rates for the Ediacaran-Cambrian transition. In the Ediacaran, there are no stage boundaries. Our work provides a framework for defining two biozones below the Ediacaran-Cambrian boundary. If we count these biozones as stages along with Fortunian stage, we have a total of three time-bins for assessing extinctions of early complex eukaryotic life. We could provide extinction rates for the terminal Ediacaran stage and the Fortunian, BUT we could not calculate a rate for the Ediacara biota biozone stage, due to the absence of older fossils.

>> Many taxa lived during two of these 'ages,' but few taxa lived in all three. So, there are relatively few 'three-timers' for calculating extinction rates. This observation may represent a consequence of incomplete sampling, or perhaps more likely, it may indicate that extinction rates were very high during the transitional times between these ages, that some clades may have been prone to rapid turnover, and that taxa were geologically short-lived during the Ediacaran-Cambrian transition. Regardless, without many 'three-timers,' it's difficult to calculate extinction rates using conventional methods.

>>> There are other problems trying to apply the concepts of 'background extinctions' and 'mass extinctions' to the Ediacaran and Cambrian.

>>> Our understanding of 'background extinction' and 'mass extinction' rates are largely based on the post-Cambrian fossil record. In general, the Cambrian is known for having the highest extinction rates of the Phanerozoic. Its 'background extinction rate' was very high, or alternatively, it witnessed multiple mass extinctions, depending on how one applies the terminology. In either case, it's difficult to compare the Cambrian to the rest of the Phanerozoic. The same problem may apply to the Ediacaran (i.e. its turnover rates may not be comparable to those of other systems).

>>> Changes in fossil preservation around Ediacaran-Cambrian transition pose another challenge for comparing rates. Ediacaran and Cambrian fossils include Ediacara-type fossils, phosphatized shelly fossils, and carbonaceous compressions (e.g. Burgess Shale-type fossils), all of which are rare in marine rocks of post-middle Cambrian age. These modes of preservation influence the sampling of Ediacaran and Cambrian fossils. In this context, Ediacaran and Cambrian extinction rates may have different degrees of sampling bias than those of younger ages of Earth history.

>>> Perhaps one day, after further sampling and additional work on the biostratigraphy, chronostratigraphy, and taphonomy of the terminal Ediacaran system, true extinction rates could be measured throughout the Ediacaran-Cambrian transition.

>>> At present, there is room to debate whether or not the episodes of biotic turnover during the Ediacaran-Cambrian transition were actually 'mass extinctions' (like the 'Big Five') or relatively minor events (e.g. 'the Botomian extinction' or the 'Serpukhovian event'). In some places, we mention that the episodes may represent the oldest 'mass extinctions' of complex life, and reference interpretations of other authors (Darroch et al., 2018; Darroch et al., 2015). We have revised the text to make it clear that these are, in fact, interpretations of the data, and do not necessarily reflect the magnitude or scale of the extinctions, relative to the events of the Phanerozoic.

Overall I found the paper to be well written but on reading it there appeared to be an unjustified jump from the results which mentioned turnover in geographic space or time as possible explanations for the distinct clusters identified and the conclusions which only mention time. Is this justifiable? If so, please make this justification clearer. If not, then the discussion, abstract and title should be modified accordingly.

>>> We have revised the text to improve the clarity of the discussion and conclusions.

>>> We do assess the role of geography by calculating assortativity coefficients for the bipartite network on Ediacaran formations and taxa. In this case, the coefficient is provided for the formation projection of that network, and measures the tendency for formations to be connected to other formations located in geographic proximity (the same country, plate, or region, for example). We received very low assortativity scores (<0.1). This result indicates that the topology of the network is not a reflection of geography. We reached similar conclusions concerning the preservational modes and

morphogroups of the taxa. By process of elimination, the best interpretation is that the clusters/module reflect associations of taxa that lived at different times.

The authors suggest that their quantitative data could form the basis for new geological stage divisions and presumably corresponding diagnostic taxa. If so, you should actually add a Table showing the break times and diagnostic taxa recovered by your analysis. I believe this would be a concrete new suggestion.

>>> We have added a new table to the text (Table 2) BUT we are not convinced that it is a worthwhile addition to the manuscript, so we leave it to the editor to decide. This information is also illustrated in the figures (Figs. 4 – 6) and described in detail in the Discussion. Also, it is available with the source data (the excel file accompanying the manuscript). So, we do not believe it is worth the space.

>> Rather than publish a large table of 185 taxa, we have limited the table to the most important taxa (i.e. best diagnostic taxa). Again, information pertaining to other taxa can be found with the source data.

>>> It is unclear to us what the reviewer means by 'break times,' but we assume this refers to the ages of the geologic boundaries between the stage-level biozones that we use in the analysis. We have included the ages of the biozones in the table.

I also attach an annotated pdf highlighting a few typos and additional minor suggestions.

>>> Thank you! We have added your comments to our tracked-changes file, and revised the text accordingly.

Reviewer #2 (Remarks to the Author):

This study takes full advantage of the possibilities of network analysis in assessing biozones among Ediacaran faunas and occurrences. The methods are state-of-the-art and the results are presented and discussed in a clear way. I think the results and also the methods are relevant for a large audience and I can recommend the publication of this manuscript largely as it is, with only two minor comments.

Lines 441-442 “was affected by two biologic turnover events, perhaps the earliest Phanerozoic-style mass extinctions.” Are there also “Precambrian-style mass extinctions”? I don’t find this term “Phanerozoic-style” very useful here – it could simply be omitted.

>>> Agreed, and we have revised the text accordingly. We borrowed the terminology of Smith et al. (2017, Proc. Roy. Soc. B) in order to keep the works consistent with each other. However, we agree that the phrase has no meaning.

Line 525-527 the measured network properties include, among others, diameter, transitivity, and edge-density. But I've not seen these three properties used or discussed anywhere in the manuscript (I didn't check the suppl. text, though); perhaps these could just be omitted from the methods description?

>>> Agreed, and we have deleted the text. We DO provide measures of these properties in the SI, but DO NOT discuss them or make use of them for interpreting the results. This revision created additional space in the methods section for a description of the sensitivity analysis.

Steffen Kiel

- Alroy, J., 2010, Fair sampling of taxonomic richness and unbiased estimation of origination and extinction rates, *in* Alroy, J., and Hunt, G., eds., *Quantitative Methods in Paleobiology*, Volume 16, *The Paleontological Society Papers*, p. 55–80.
- Alroy, J., 2015, A more precise speciation and extinction rate estimator: *Paleobiology*, v. 41, no. 4, p. 633–639.
- Boag, T. H., Darroch, S. A. F., and Laflamme, M., 2016, Ediacaran distributions in space and time: Testing assemblage concepts of earliest macroscopic body fossils: *Paleobiology*, v. 42, no. 4, p. 574–594.
- Darroch, S. A. F., Smith, E. F., Laflamme, M., and Erwin, D. H., 2018, Ediacaran Extinction and Cambrian Explosion: *Trends in Ecology & Evolution*, v. 33, no. 9, p. 653–663.
- Darroch, S. A. F., Sperling, E. A., Boag, T. H., Racicot, R. A., Mason, S. J., Morgan, A. S., Tweedt, S., Myrow, P., Johnston, D. T., Erwin, D. H., and Laflamme, M., 2015, Biotic replacement and mass extinction of the Ediacara biota: *Proceedings of the Royal Society of London B: Biological Sciences*, v. 282, no. 1814, p. 1–10.
- Foote, M., 2000, Origination and extinction components of taxonomic diversity: Paleozoic and post-Paleozoic dynamics: *Paleobiology*, v. 26, no. 4, p. 578–605.
- Gehling, J. G., and Droser, M. L., 2013, How well do fossil assemblages of the Ediacara Biota tell time?: *Geology*, v. 41, no. 4, p. 447–450.
- Grazhdankin, D., 2004, Patterns of distribution in the Ediacaran biotas: facies versus biogeography and evolution: *Paleobiology*, v. 30, no. 2, p. 203–221.
- Hagadorn, J. W., Fedo, C. M., and Waggoner, B. M., 2000, Early Cambrian Ediacaran-type fossils from California: *Journal of Paleontology*, v. 74, no. 4, p. 731–740.
- Shen, B., Dong, L., Xiao, S., and Kowalewski, M., 2008, The Avalon explosion: Evolution of Ediacara morphospace: *Science*, v. 319, p. 81–84.
- Smith, E. F., Nelson, L. L., Strange, M. A., Eyster, A. E., Rowland, S. M., Schrag, D. P., and Macdonald, F. A., 2016, The end of the Ediacaran: Two exceptionally preserved body fossil assemblages from Mount Dunfee, Nevada, USA: *Geology*, v. 44, p. 911–914.
- Smith, E. F., Nelson, L. L., Tweedt, S. M., Zeng, H., and Workman, J. B., 2017, A cosmopolitan late Ediacaran biotic assemblage: new fossils from Nevada and Namibia support a global biostratigraphic link: *Proceedings of the Royal Society B: Biological Sciences*, v. 284, no. 1858, p. 20170934.

Waggoner, B., 1999, Biogeographic analyses of the Ediacara biota; a conflict with paleotectonic reconstructions: *Paleobiology*, v. 25, no. 4, p. 440-458.

Waggoner, B., 2003, The Ediacaran biotas in space and time: *Integrative and Comparative Biology*, v. 43, no. 1, p. 104–113.

Reviewers' Comments:

Reviewer #1:

Remarks to the Author:

I think the authors have done a good job in revising the manuscript. They have taken reviewer comments on board and responded constructively including by adding additional statistical analyses as suggested. Therefore I recommend publication with only minor revisions.

I would suggest that the authors add (or move) a sentence to the start of discussion paragraphs 2-3 to briefly summarise the main message of the paragraph before they launch into technical discussions.

I think Table 2 is a useful addition as it makes explicit the specific diagnostic taxa the study associates with the hypothesised biozones. This is, of course, likely to have more impact if it remains in the main text but I agree it could be moved to the supplementary material if space is short.

REVIEWERS' COMMENTS:

Reviewer #1 (Remarks to the Author):

I think the authors have done a good job in revising the manuscript. They have taken reviewer comments on board and responded constructively including by adding additional statistical analyses as suggested. Therefore I recommend publication with only minor revisions.

>> Thank you for taking the time to review our manuscript a second time!

I would suggest that the authors add (or move) a sentence to the start of discussion paragraphs 2-3 to briefly summarise the main message of the paragraph before they launch into technical discussions.

>> We have revised the text, so that the main messages of these paragraphs will now be clear.

I think Table 2 is a useful addition as it makes explicit the specific diagnostic taxa the study associates with the hypothesised biozones. This is, of course, likely to have more impact if it remains in the main text but I agree it could be moved to the supplementary material if space is short.

>> Agreed. If space is not an issue, we would prefer to see the table published along with the main text.

Dr J Hoyal Cuthill